# LLMDet: A Third Party Large Language Models Generated Text Detection Tool

**Kangxi Wu**[1,2], **Liang Pang**[1]*, **Huawei Shen**[1,2], **Xueqi Cheng**[1,2], **Tat-Seng Chua**[3]

[1] Institute of Computing Technology, Chinese Academy of Sciences
[2] University of Chinese Academy of Sciences
[3] Sea-NExT Joint Lab, National University of Singapore
{wukangxi22s, pangliang, shenhuawei, cxq}@ict.ac.cn
dcscts@nus.edu.sg

## Abstract

Generated texts from large language models (LLMs) are remarkably close to high-quality human-authored text, raising concerns about their potential misuse in spreading false information and academic misconduct. Consequently, there is an urgent need for a highly practical detection tool capable of accurately identifying the source of a given text. However, existing detection tools typically rely on access to LLMs and can only differentiate between machine-generated and human-authored text, failing to meet the requirements of fine-grained tracing, intermediary judgment, and rapid detection. Therefore, we propose LLMDet, a model-specific, secure, efficient, and extendable detection tool, that can source text from specific LLMs, such as GPT-2, OPT, LLaMA, and others. In LLMDet, we record the next-token probabilities of salient $n$-gram as features to calculate proxy perplexity for each LLM. By jointly analyzing the proxy perplexities of LLMs, we can determine the source of the generated text. Experimental results show that LLMDet yields impressive detection performance while ensuring speed and security, achieving 98.54% precision and about $\times 5.0$ faster for recognizing human-authored text. Additionally, LLMDet can effortlessly extend its detection capabilities to a new open-source model. We will provide an open-source tool at https://github.com/TrustedLLM/LLMDet.

## 1 Introduction

Recently, the emergence of ChatGPT[1] has heralded a "Cambrian Explosion" for generative large language models (LLMs). GPT-4 (OpenAI, 2023), Bard[2], PaLM-2 (Anil et al., 2023), and other LLMs from internet companies are currently flourishing, while open-source communities are witnessing a proliferation of open-source models like LLaMA (Touvron et al., 2023a), OPT (Liu et al., 2021), ChatGLM (Du et al., 2022). These models are capable of generating coherent, fluent, and meaningful text. However, the formidable text generation capabilities of generative language models have also raised concerns about their potential misuse in domains such as phishing, spreading false information, and academic fraud. Additionally, with the application of products like ChatGPT, the future abundance of machine-generated text data has the potential to contaminate genuine human-generated data (Hataya et al., 2022), altering the data ecosystem of the real world.

Accordingly, the study of practical content generation detection tools has attracted widespread attention from the community. Recently, the primary focus of research is on approaching the text detection problem as a binary classification task to distinguish machine-generated text and human-authored text, making it hard to assign responsibility to a specific model or its provider. Nevertheless, Watermarking (Kirchenbauer et al., 2023) methods necessitate altering the text generation process, leading to a compromise in the quality of the generated content. Techniques like GPT-zero[3], Detect-GPT (Mitchell et al., 2023), and the classifier in OpenAI (OpenAI, 2023) require access to the deployed model, thereby resulting in high cost and intractability for third parties.

Thus, a practical LLM detection tool should possess the following capabilities, which are also the objectives of our method: **Specificity**: Merely focusing on identifying human and machine-generated text is insufficient for duty attribution. There is a pressing need for the ability to recognize the specific model responsible for generating the text. **Safety**: Ensuring model security and mitigating potential risks require a detection method that does not require accessing model parameters. This need is particularly urgent for commercial mod-

---

*Corresponding author
[1]https://openai.com/product/chatgpt
[2]https://bard.google.com
[3]https://gptzero.me

els. **Efficiency**: With the increasing demand for detection and the exponential growth of models, it is crucial to develop detection algorithms that have low resource and low latency requirements. **Extendibility**: The detection tool should inherently possess the capacity to seamlessly accommodate emerging model paradigms. This capability plays a pivotal role in refining the detection ecosystem and effectively addressing the ever-expanding variety of LLMs.

Guided by the aforementioned capabilities, we propose a pragmatic third-party detection method called LLMDet. Our approach is inspired by the observation that perplexity serves as a reliable signal for distinguishing the source of generated text, a finding that has been validated in previous work (Solaiman et al., 2019; Jansen et al., 2022; Mitchell et al., 2023). However, directly calculating perplexity requires access to LLMs, which compromises both safety and efficiency. In LLMDet, we address this challenge by capturing the next token probabilities of prominent $n$-gram in texts as priors. This enables us to efficiently compute a proxy perplexity for each LLM. By comprehensively analyzing the proxy perplexities of LLMs, we can accurately trace the specific language model responsible for generating the text. Notably, our method eliminates the need to access the model at the detection end, ensuring the security of parameters in large-scale models. It also offers the potential for seamless integration with emerging open-source models, as well as proprietary models under appropriate licensing. These factors contribute to the widespread adoption of our approach.

LLMDet exhibits outstanding overall detection performance, with an F1-Macro score of 88.14% and near-perfect results for R@2, indicating that highly ranked predictions cover the correct labels for the majority of instances. Particularly notable is its exceptional discriminative ability in human text, LLaMA-generated text, and BART-generated text. In terms of detection efficiency, LLMDet significantly outperforms other similar methods such as fine-tuned RoBERTa, GPT-zero[4], Detect-GPT (Mitchell et al., 2023), and True-PPL with respect to speed. And, it has very low resource requirements, as text detection can be accomplished solely on a CPU, enabling easy accessibility for a wider range of users. Additionally, when tested on perturbated text data, LLMDet produces satisfac-

tory detection results, demonstrating its robustness and adaptability.

## 2 Related Work

The existing methods for detecting generated text can be broadly categorized into two types: black-box and white-box detection (Tang et al., 2023).

### 2.1 Black-box Detection

Black-box detection methods can be further divided into three main branches: statistical learning methods, supervised learning methods, and unsupervised learning methods. Traditional approaches utilize statistical metrics such as entropy, perplexity, and $n$-gram frequency for text classification (Gehrmann et al., 2019; Fröhling and Zubiaga, 2021).

Compared to statistical learning methods, supervised learning methods are more commonly used in text detection. These works leverage text features to train a supervised classification model specifically designed for the detection of machine-generated text (Bakhtin et al., 2019; Uchendu et al., 2020; Fagni et al., 2021; OpenAI, 2023).

However, the study conducted by (Uchendu et al., 2020; Chakraborty et al., 2023) demonstrates that a limitation of supervised models is the potential occurrence of overfitting within the domain, resulting in poor detection performance outside the domain.

To address the limitations of supervised learning methods, unsupervised learning methods such as DetectGPT (Mitchell et al., 2023) and GPT-Zero have been developed. These approaches utilize checks on perplexity and burstiness in the text to determine whether it is artificially generated or authored by a human.

### 2.2 White-box Detection

White-box detection methods require full access to LLMs, thereby enabling control over the generation behavior of the model or embedding watermark within the generated text (Abdelnabi and Fritz, 2021; Ueoka et al., 2021; Dai et al., 2022). This enables the tracking and detection of machine-generated text within white-box settings.

The current state-of-the-art approach, as proposed by (Kirchenbauer et al., 2023), partitions the model's vocabulary into whitelist and blacklist tokens when predicting the next token given a prompt. During text generation, the goal is to

---

[4]https://gptzero.me

produce whitelist tokens as much as possible, effectively creating a strong watermark. Third parties can determine if the text is machine-generated by analyzing the frequency of whitelist tokens within the text. While watermarking methods offer robustness and interpretability, they can compromise the quality of the generated text and may not be highly practical in certain scenarios (Sadasivan et al., 2023).

## 3 Motivation

A practical LLMs detection method should possess the characteristics of being specific, secure, efficient, and extensible, which serve as the intention for developing our third-party detection tool.

**Specificity**: The field of LLMs constantly evolves, indicating that a sole focus on identifying human and machine-generated text is insufficient to meet detection requirements. From the perspective of copyright protection for works generated by artificial intelligence (Aplin and Pasqualetto, 2019), an ideal detection tool should be capable of identifying the specific language model responsible for generating the text, thereby exerting a lasting impact on intellectual property rights protection.

**Safety**: The majority of existing detection methods require accessing or modifying model parameters, which is deemed unacceptable for commercial models. Once the model is loaded, it represents a financial loss for the owner and can also expose the model to potential attacks (Kurita et al., 2020). Hence, considering the security of the model, it is desirable to minimize the need for model loading during the detection process.

**Efficiency**: With the growing number of users utilizing large-scale models, the future of text detection is poised for exponential expansion in terms of demand and user base. For instance, in the realm of education, there is a significant need for text detection to combat cheating and plagiarism (Cotton et al.), despite often constrained hardware conditions. This poses a formidable challenge to existing detection methods. Hence, the pursuit of rapid and resource-efficient approaches has become a pivotal direction in developing efficient detection algorithms.

**Extendibility**: As for multi-model generated text detection approaches, it is crucial to seamlessly adapt to emerging model paradigms and extend detection capabilities to new models. This is because an excellent detection tool is not static but needs to keep up with technological advancements and continuously enhance its own detection ecosystem to address the challenges posed by new models.

## 4 LLMDet

Combining the aforementioned motivations, we introduce LLMDet, a text detection tool capable of identifying the sources from which the text was generated, such as Human, LLaMA, OPT, or others. The overall framework of the system is illustrated in Figure 1 and consists of two main components: Dictionary Construction (see § 4.1) and Text Detection (see § 4.2).

The construction of the dictionary is performed offline by us or provided by the model owner, ensuring its independence from external systems. This ensures the fulfillment of the four characteristics proposed for our detection tool in § 3. The text detection component can be distributed to tool users, allowing third-party detection without requiring the possession of the model. For the specific algorithm, please refer to Appendix A.

### 4.1 Dictionary Construction

Drawing from previous detection works, such as DetectGPT (Mitchell et al., 2023) and GPT-Zero[5], perplexity has shown promising results in detecting machine-generated text. Therefore, we consider utilizing perplexity as a measurement of identifying the generated text from different LLMs. However, calculating the actual perplexity requires access to LLMs, which goes against the safety and efficiency characteristics of the practical LLMs detection method.

Perplexity is a measure used to evaluate the performance of language models. Specifically, it is the exponential average of the negative log-likelihood of a sequence generated by the model. The perplexity score is calculated based on the probability of generating the next word, given all the previous words in the sequence, e.g. $p(x_i, x_{<i})$. In order to calculate the perplexity of text without accessing the model, we need approximate $p(x_i, x_{<i})$ by replacing $x_{<i}$ with a $n$-gram , thus a dictionary should be constructed, with $n$-gram as keys and the next token probabilities as values. This dictionary serves as prior information during the detection process, allowing us to compute the proxy perplexity of the text instead of the true perplexity. The construction process can be divided into three steps:

---

[5]https://gptzero.me

Figure 1: The detailed processes of the proposed tool LLMDet. It contains two main phases, dictionary construction and text detection. The dictionary construction phase is carried out offline by us or provided by the model holder, independent of external systems. The text detection phase can be accessed by the tool user who, as a third party, performs text detection without holding the model.

**1) Generated Text Sampling**: Due to the absence of readily available model-generated text data, it is necessary to collect a sufficient number of corresponding generated texts for each model. We provide a prompt dataset and, for each model, randomly sample an equal number of prompts. We use these prompts to generate corresponding texts and collect the required text data.

**2) Word Frequency Statistics**: In this phase, we first utilize the generated texts collected in the previous step to perform $n$-gram word frequency statistics (Pang et al., 2016). The $n$-gram range from 2-gram to $n$-gram. Subsequently, we select the top-$k$ $n$-gram based on their frequency.

**3) Next Token Probability Sampling**: In this phase, we use each $n$-gram $s$ obtained from word frequency statistics as samples. We input the first $n-1$ token $s_{[1:n-1]}$ into the corresponding generative models for predicting next-token probabilities $p^w = [p_1^w, \ldots, p_{|\mathcal{W}|}^w]$, where $|\mathcal{W}|$ is the size of vocabulary. Subsequently, we sample the top-$K$ words based on next-token probabilities. For $n$-gram with different values of $n$, the optimal value of $K$ for top-$K$ sampling may vary.

We should consider the optimal values, the degree of $n$-gram, the number of $n$-gram $k$, and the number of next token probabilities $K$ from two aspects: detection performance and storage cost.

In terms of detection performance, the larger $n$, $k$, and $K$ may improve the detection performance of LLMDet, as this enables the proxy perplexity to approximate the true perplexity.

In terms of storage cost, due to the data type of the sampling probabilities being Float64 and $n$-gram being a string, a significant amount of storage space is required, e.g. $O(nkK)$. If $n$ is set to 4, $k$ is set to 100,000 (much smaller than the number of 4-gram), and $K$ is set to 10,000 (most vocabulary size is larger than that), we need almost 22GB to store only probabilities for one model. Thus, we have to reduce the storage in practical use. The reduction can be considered in two folds, 1) select a suitable $n$, $k$ and $K$, 2) reduce Float64 to Float16 and represent $n$-gram as Int16. We find that does not significantly affect LLMDet, while it reduces storage costs by approximately 11 times about 0.5GB.

In the end, we constructed an $n$-gram and probability dictionary for each LLM, which was utilized for calculating proxy perplexity. The above three steps are repeated on GPT-2 (Radford et al., 2019), OPT (Liu et al., 2021), UniLM (Dong et al., 2019), LLaMA (Touvron et al., 2023a), BART (Lewis et al., 2019), T5 (Raffel et al., 2020), Bloom (Scao et al., 2022) and GPT-neo (Black et al., 2022), respectively.

## 4.2 Text Detection

In § 4.1, we have obtained the dictionary of $n$-gram and their probabilities. Therefore, we can use the corresponding dictionary of each model as prior information for third-party detection to calculate the proxy perplexity of the text being detected on each model. Immediately after, by inputting the proxy perplexity as a feature into a trained text classifier, we can obtain the corresponding detection results.

### 4.2.1 Proxy Perplexity Estimating

During text detection, for the input text $X$, our initial task is to estimate the proxy perplexity of

this text across various large language models as a vector of feature information.

Taking the estimation of proxy perplexity on $Model_m$ as an example, we begin by tokenizing the input text $X$ to obtain its sequence $X = [x_1, x_2, ..., x_t]$, assuming the length of the tokenized sequence is denoted as $t$.

Then, the proxy perplexity of the sequence $X$ on $Model_m$ can be mathematically represented by the following function, denoted as Proxy_PPL:

$$\text{Proxy\_PPL}(X) = -\frac{1}{t} \sum_{i=0}^{t} \log p\left(x_i \mid n\text{-gram}\right). \quad (1)$$

More specifically, $\log p\left(x_i \mid n\text{-gram}\right)$ represents the logarithmic likelihood of the i-th token, conditioned on the preceding tokens $x_{<i}$ matching the $n$-gram in the dictionary of $Model_m$. The likelihood probability $p\left(x_i \mid n\text{-gram}\right)$ corresponds to the value associated with the matching $n$-gram in the dictionary.

Similarly, by repeating the above procedure on other models, we can obtain the proxy perplexity of the detection text on the respective models. These proxy perplexities constitute the feature information vector for detection, denoted as $\mathbf{F} = [\text{Proxy\_PPL}_1, \text{Proxy\_PPL}_2, ..., \text{Proxy\_PPL}_c]$, subscript $c$ denotes the number of LLMs.

#### 4.2.2 Result Ranking

Before result ranking, we initially estimate the proxy perplexity of the generated texts from each language model and human-generated texts. This estimation allows us to obtain a separate feature information vector for each text. Subsequently, these vectors are employed to train a text detector.

Next, we input the feature information vectors $\mathbf{F}$, obtained during the proxy perplexity estimation phase, of the text to be detected into the trained text detector for result prediction, yielding a prediction result, such as for a given $Model_i$, the probability is denoted as $p_i$. It is important to note that we denote the probability of $Human$ as $p_0$.

However, due to the fact that the text detector is trained based on perplexity as a feature, it is not sensitive to the length information of the detected text, resulting in suboptimal detection performance for some short texts. Therefore, it is necessary to apply a smoothing technique to the probabilities of the detection results in order to enhance the success rate of detecting short texts. The smoothing process

is denoted as,

$$\tilde{p}_i = \log\left(p_i\right) + \frac{1}{L} \log\left(\frac{1}{c+1}\right), \quad (2)$$

with $L$ is the length of the text to be detected, $c$ denotes the number of LLMs.

Finally, we apply softmax to the smoothed probabilities to obtain $[\hat{p_0}, \hat{p_1}, ..., \hat{p_c}]$. Consequently, the detection results are transformed into the probability of $Model_i$ is $\hat{p_i}$. Subsequently, the detection results are sorted based on the magnitude of the probability values in the result dictionary, yielding the final detection outcome,

$$[\hat{p_0}, \hat{p_1}, ..., \hat{p_c}] = \text{softmax}\left([\tilde{p_0}, \tilde{p_1}, ..., \tilde{p_c}]\right). \quad (3)$$

## 5 Experiments

We conduct experiments based on proxy perplexity and true perplexity according to the methods proposed in § 4. By comparing the performance of the text detectors based on fine-tuned RoBERTa, proxy perplexity, and ture perplexity, we find that our proposed method outperforms existing methods in terms of detection efficiency, security, and scalability while ensuring the performance of the detector.

### 5.1 Datasets

In our experiments, we use Wikipedia paragraphs from the SQuAD context (Rajpurkar et al., 2016) and news articles from the Xsum (Narayan et al., 2018) dataset for extraction. We extract the first 5 phrases of each text data to form a prompt dataset. During the text generation phase, for each LLM, we randomly select 32,000 data samples from the prompt dataset as input and have the model generate corresponding text. The generated text from each model is evenly split into two parts: 16,000 samples for the statistical dataset and 16,000 samples for the validation dataset. The statistical dataset is used for $n$-gram frequency counting. The validation dataset from LLMs, along with 16,000 samples collected from HC3 (Guo et al., 2023) as human-generated text, form a combined dataset for the training and validation of text detectors.

### 5.2 Metrics

To evaluate the ability of the detector to distinguish between text generated by different LLMs and human-written text, we employ precision ($P$), recall ($R$), and F1 score to assess the discriminative

Table 1: Experimental results of text detector based on FT-RoBERTa(Fine-tuned RoBERTa), True-PPL(True Perplexity), and LLMDet(Proxy Perplexity). Their detection environments are respectively GPU-V100 32GB, CPU, and CPU.

| Metric | Method | Label of Text Source | | | | | | | | |
|--------|--------|-------|-------|------|-------|-------|--------|-------|-------|---------|
| | | Human | GPT-2 | OPT | UniLM | LLaMA | BART | T5 | BLOOM | GPT-Neo |
| P(%) ↑ | FT-RoBERTa | 90.82 | 85.61 | 53.35 | 91.67 | 44.62 | 100.00 | 73.95 | 70.80 | 42.08 |
| | True-PPL | 97.97 | **98.54** | **98.25** | 79.02 | **98.54** | **98.94** | **89.77** | **94.41** | **97.09** |
| | LLMDet | **98.54** | 76.09 | 79.08 | **90.81** | 95.61 | 97.55 | 86.86 | 84.67 | 84.45 |
| R(%) ↑ | FT-RoBERTa | 94.00 | 58.05 | 81.48 | 73.41 | 95.24 | 94.18 | 27.98 | 27.09 | 18.22 |
| | True-PPL | 98.99 | **95.92** | **95.70** | 82.25 | **98.46** | **99.73** | **88.44** | **94.29** | **97.61** |
| | LLMDet | **99.00** | 78.13 | 73.88 | **91.74** | 97.30 | 98.41 | 87.56 | 83.08 | 83.90 |
| F1(%) ↑ | FT-RoBERTa | 92.38 | 69.19 | 64.48 | 81.53 | 60.77 | 97.00 | 40.60 | 39.19 | 25.13 |
| | True-PPL | 98.48 | **97.22** | **96.96** | 80.60 | **98.85** | **99.34** | **89.10** | **94.35** | **97.35** |
| | LLMDet | **98.77** | 77.09 | 76.39 | **91.27** | 96.44 | 97.98 | 87.21 | 83.87 | 84.18 |

Table 2: Comparison of overall performance between text detectors based on true perplexity, fine-tuned RoBERTa, and proxy perplexity. Note: $Ratio = (Macro\text{-}F1/Macro\text{-}F1_{True\_PPL}) \cdot (Time/Time_{Ture\_PPL})$

| Method | Macro-P(%) ↑ | Macro-R(%) ↑ | Macro-F1(%) ↑ | Time(s) ↓ | Ratio ↑ |
|--------|--------------|--------------|---------------|-----------|---------|
| True-PPL | 94.72 | 94.60 | 94.65 | 46410.15 | ×1.00 |
| Fine-tuned RoBERTa | 72.19 | 63.30 | 63.40 | 41799.76 | ×0.74 |
| LLMDet | 88.19 | 88.13 | 88.14 | 8678.76 | **×4.97** |

performance of the text detector on each of LLMs and human-generated text. Additionally, F1-Macro, R@1, R@2, and R@3 metrics are used to analyze the overall performance of the detector,

$$F1_i = \frac{2P_i R_i}{P_i + R_i}, \quad F1\text{-Macro} = \frac{\sum_{i=1}^{N} F1_i}{N}, \quad (4)$$

$$R@k = \frac{\sum_{j=1}^{M} \mathbb{I}_{G_j \in K_j}}{M}, \quad (5)$$

where $P_i$, $R_i$ and $F1_i$ respectively represent the precision, recall, and F1 score of $Model_i$. $N$ denotes the total number of categories, $M$ represents the number of texts being tested. $G_j$ represents the ground label of Text $j$, $K_j$ refers to the top-$k$ categories with the highest probabilities in the predicted results, $\mathbb{I}_{G_j \in K_j}$ takes the value of 1 when $G_j \in K_j$, and 0 otherwise.

## 5.3 Research Quesitons

Based on the characteristics and assumptions of our proposed detection tool in § 3, we formulate four research questions regarding LLMDet.

- **RQ1**: Can perplexity-based methods trace the source of text from certain LLM?

- **RQ2**: How significant is the impact of the proxy perplexity-based approach on detection performance?

- **RQ3**: Can LLMDet achieve the expected level of efficiency compared to existing methods?

- **RQ4**: How is the extendibility of LLMDet demonstrated?

## 5.4 Experiments & Results

We conducted experimental verification for the aforementioned raised questions.

**For Specificity (RQ1)**: We first compute the true perplexity of the combined datasets constructed in § 5.1 on GPT-2, GPT-2-Large, OPT-1.3B, OPT-2.7B, UniLM, LLaMA-7B, BART, T5-Base, Bloom-650M and GPT-Neo-2.7B models. Subsequently, we joint these perplexity values to train a text classifier based on LightGBM (Ke et al., 2017).

The classifier is then tested, and the results are presented in Table 1. We observe that the text detector based on true perplexity achieved excellent detection success rates when confronted with texts generated by different models, with the exception of the generated texts by UniLM. Despite

the comparatively lower detection performance for UniLM-generated texts, the F1 score reaches 80.60%, which is significantly higher than random guessing. These experimental results robustly validate the applicability of perplexity as a distinguishing metric for models that identify specific sources of text.

**For Safety (RQ2)**: We utilize the statistical datasets generated on GPT-2, GPT-2-Large, OPT-1.3B, OPT-2.7B, UniLM, LLaMA-7B, BART, T5-Base, Bloom-650M, and GPT-Neo-2.7B, as mentioned in the § 5.1, to construct dictionaries for each model using the method described in the § 4.1. Then, we employ these dictionaries to calculate the proxy perplexity of the combined dataset as features for training a text classifier based on Light-GBM (Ke et al., 2017).

The classifier is then tested, and the results are presented in Table 1. Our proposed method based on proxy perplexity achieves comparable results to the text detector based on real perplexity on Human, LLaMA-generated, and BART-generated texts, with detection success rates exceeding 95%. Additionally, our method outperforms the true perplexity-based detector when it comes to detecting UniLM-generated texts. Furthermore, the F1 score for detecting texts from other sources is at least 76.39%, significantly higher than random guessing. Based on the confusion matrix in Figure 2, it can be observed that there is a tendency for the text generated by GPT-2 and OPT to be easily confused with each other, while text generated by T5, Bloom, and GPT-Neo also exhibit a tendency to be easily confused. Although the overall performance is not as high as the real perplexity-based text classifier, our proposed method does not require model access during detection and offers advantages such as speed, scalability, and enhanced security.

To assess the comprehensive detection capability of the detector, we compute the F1-Macro, R@1, R@2 and R@3 values. From Table 2, it is evident that our proposed method achieves an R@2 value of 98.00%. This indicates that, among the top two text sources with the highest predicted probabilities, there is typically one source that corresponds to the true source of the text.

**For Efficiency (RQ3)**: In order to compare the efficiency of various methods, in addition to the main experiment in Table 2, we also conduct tests using the same set of 1000 texts to measure the efficiency required for detection in GPT-Zero, De-

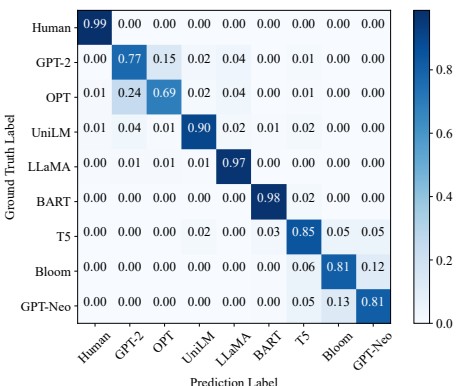

Figure 2: The confusion matrix of the detection performed by LLMDet.

Table 3: The detection time of GPT-Zero, DetectGPT, and LLMDet on a dataset of 1000 texts. Note: $Ratio = (Accuracy/Accuracy_{True\_PPL}) \cdot (Time/Time_{Ture\_PPL})$

| Method | Accuracy(%) ↑ | Time(s) ↓ | Ratio (True-PPL) ↑ |
|---|---|---|---|
| GPT-Zero | 86.56 | 2376.87 | ×0.46 |
| DetectGPT | 92.67 | 14354.61 | ×0.08 |
| True-PPL | **94.87** | 1199.11 | ×1.00 |
| LLMDet | 88.19 | **224.53** | **×4.96** |

tectGPT, True-PPL, and LLMDet. In terms of resource requirements, both DetectGPT and True-PPL methods are run on a V100-SXM-32GB, GPT-Zero utilizes its API for detection on a GPU, while LLMDet only requires a GPU for the completion of the detection process.

Based on the efficiency analysis in Table 2 and Table 3, it can be observed that LLMDet outperforms other detection methods significantly. Furthermore, in terms of resource requirements, our approach exhibits the lowest demands. Consequently, our detection tool demonstrates a substantially higher efficiency compared to other methods, making it more aligned with future detection needs.

**For Extendibility (RQ4)**: To illustrate the extendibility of the LLMDet method, we expand its detection capability from one model to eight. Specifically, We sequentially add the LLM model into our LLMDet tool in the following sequence: GPT-2, LLaMA, OPT, UniLM, BART, T5, Bloom, and GPT-Neo. Thereby, continuously extending the detection capability to these models. Additionally, with each expansion, we retrain the text detector (LightGBM) and assessed the resultant changes in overall performance.

From Figure 3, it can be observed that during the expansion of LLMDet, there is only a slight fluctuation in the value of F1-Macro, which remains

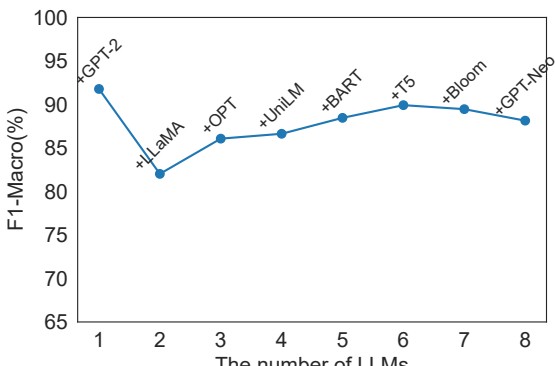

Figure 3: The impact of sequentially adding the LLM into LLMDet on the comprehensive detection performance measured by F1-Macro.

consistently around 85%. Therefore, it can be concluded that in the future, LLMDet can be easily expanded to a new model with sightly performance affection.

In addition, in order to explore the performance changes of LLMDet when using newer and larger LLM, we also conducted additional experiments. The detailed experimental steps and results can be seen in Appendix B.

## 6 Analysis

In this section, we conduct several additional experiments to facilitate a more comprehensive analysis of LLMDet. Firstly, we verify the detection robustness of LLMDet. Subsequently, we investigate the impact of $n$-gram in dictionary construction on the detection performance of LLMDet. Finally, we explore the influence of the top-$k$ of the next token samples in dictionary construction on the detection performance of LLMDet.

### 6.1 The Robustness Testing of Detector

Many LLMs can change their probability of the next token via different methods, for example, changing hyperparameters like temperature, or even updating weight by fine-tuning. Furthermore, generated text may encounter deliberate perturbation, such as random deletions. It is worth considering the robustness of this method in these situations.

For hyperparameter changes, we use the approach outlined in § 5.1 of this article to generate 16,000 text instances using LLaMA-7B at temperatures of 0.1, 0.4, 0.7, and 1.0 respectively.

For random deletion, we use the approach out-

lined in § 5.1 to generate 16,000 text instances using LLaMA-7B. For the generated text, we set the deletion rates at 0.1, 0.3, and 0.5, respectively, subsequently introducing corresponding perturbed texts by randomly removing words from the text according to these specified rates.

For weight updates, we employ the approach outlined in § 5.1 to generate 16,000 text instances using the Vicuna-7B, an instruction fine-tuned version of LLaMA-7B.

These text instances are then utilized as test data to assess the robustness of LLMDet, and the experimental outcomes are presented in Table 4. LLMDet exhibits strong robustness against certain types of perturbations in the text, such as random deletions, slight weight updates in the generative model, and adjustments to temperature settings. For more analysis of experimental results, please see Appendix C.

### 6.2 The Influence of $N$-gram

We compute the proxy perplexity of each model for the combined dataset in the § 4.1 using dictionaries built on 2-gram, 3-gram, and 4-gram, respectively. By jointly analyzing the proxy perplexities to train and test the text classifier using the LightGBM. It should be noted that $(n$-1$)$-gram are a subset of $n$-gram. Based on the results shown in Table 5, it can be observed that the overall detection performance of text within the domain does not increase significantly as the value of $n$ increases, but rather exhibits a slight improvement. Considering that the number of $n$-gram increases exponentially as $n$ increases, we only consider 4-gram in LLMDet.

### 6.3 Next Token Top-$K$ Sampling

The construction of the dictionary incurs significant storage overhead due to the necessity of storing the top-$K$ probabilities along with their corresponding $n$-gram, presenting a challenge to our method. Consequently, determining the optimal value of $K$ requires a comprehensive consideration of both detection performance and storage costs.

In order to gain a more intuitive understanding of the impact of the $K$ value on the detection performance of LLMDet, while keeping the number of 2-gram, we solely vary the $K$ value and examine the changes in F1-Macro of LLMDet across different $K$ values. The result is presented in Figure 4.

We observe that as the value of $K$ increases, the detection performance of LLMDet gradually improves. However, the performance improvement

Table 4: The detection performance of LLMDet in three scenarios: temperature changes, random deletion, and weight updates.

| Metric (%) | Temperature | | | | Delete Ratio | | | Weight Update |
| --- | --- | --- | --- | --- | --- | --- | --- | --- |
| | 0.1 | 0.4 | 0.7 | 1.0 | 0.1 | 0.3 | 0.5 | Fine-tuned LLaMA(Vicuna) |
| R@1(Accuracy) ↑ | 91.23 | 92.05 | 93.02 | 94.37 | 90.06 | 89.31 | 87.80 | 97.78 |
| R@2 ↑ | 97.55 | 97.48 | 97.48 | 98.06 | 97.07 | 99.12 | 99.53 | 99.07 |
| R@3 ↑ | 99.46 | 99.33 | 99.24 | 99.33 | 99.52 | 99.53 | 99.82 | 99.39 |

Table 5: The impact of the value of $n$ in $n$-gram on the overall detection performance.

| Metric (%) | 2-gram | 3-gram | 4-gram |
| --- | --- | --- | --- |
| F1-Macro ↑ | 87.44 | 87.79 | 88.14 |
| R@1 ↑ | 89.61 | 90.18 | 89.51 |
| R@2 ↑ | 97.84 | 98.04 | 98.00 |
| R@3 ↑ | 99.58 | 99.56 | 99.64 |

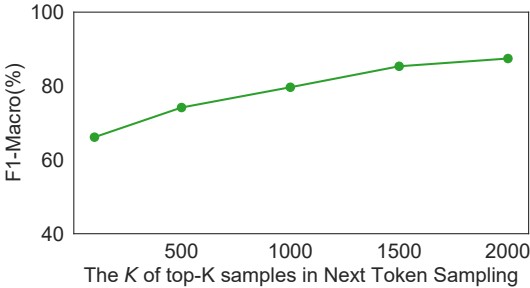

Figure 4: The impact of the $K$ value in top-$K$ sampling of 2-gram on the detection performance of LLMDet.

becomes less pronounced after $K$ reaches 1500. Nonetheless, the corresponding storage overhead still increases linearly. Therefore, considering the overall trade-off between detection performance and storage cost, we recommend adopting a top-2000 sampling for 2-gram. For 3-gram and 4-gram, their quantities are immense. Therefore, following the completion of similar experimental analyses, we employ a top-100 sampling for these $n$-gram .

## 7 Conclusions and Future Work

In the era dominated by machine-generated text, there is a growing need for an efficient and secure detection tool. However, existing detection methods typically require interaction with language models, which inherently compromises speed and security. Our proposed detection tool, LLMDet, overcomes these limitations by leveraging pre-mined prior probability information to compute proxy perplexity, ensuring both speed and secu-

rity in the detection process. Additionally, our method enables text tracking, allowing for the identification of the underlying language model from which the text originates. Importantly, our detection tool can be continuously enhanced by expanding to new open-source LLMs, enabling ongoing improvements.

In the future, we aim to further refine our detection tool. Firstly, we will improve the dictionaries used to compute proxy perplexity, thereby enhancing the detection performance. Secondly, for closed-source models, we are unable to build their corresponding dictionaries. To mitigate it to some extent, we have considered two possible approaches:

**1)** In the process of implementing LLMDet, we offer not only detection capabilities but also an extensible interface for closed-source model owners. Details about this implementation can be found in Algorithm 1 of Appendix A. The extended interface aims to secure the model effectively without compromising the interests of the model owners. Through this approach, we hope to encourage more closed-source model owners to participate and contribute to the continuous improvement of the detection ecosystem of LLMDet.

**2)** We have also explored using statistical techniques to estimate the next-token probability in proprietary commercial models. However, due to limited data volume, achieving the anticipated results has been challenging. Additionally, generating a significant amount of statistical data comes with considerable costs. As a result, we have included this approach on our list of future work items.

Furthermore, the distillation method is a valuable avenue for future exploration. We will certainly consider it in our future research endeavors.

## Limitations

One of the limitations of the current LLMDet is its restriction to detecting English text, thus unable to detect text in other languages. In the future, we can

extend our approach to encompass models for other languages, thereby equipping it with the capability to detect text in diverse languages.

Furthermore, at present, the number of models detectable by LLMDet is limited. We will expand the capabilities of our detection tool to encompass a broader range of models, providing more possibilities for text tracing and attribution.

## Ethics Statement

We honor and support the ethical guidelines of EMNLP. This paper primarily focuses on the detection of text generated by LLMs, aiming to construct a detection tool suitable for the user base from various domains. The tool is designed to efficiently and securely perform text detection to prevent the misuse of generated text. Overall, our approach exhibits advantages over previous methods in terms of efficiency and granularity of detection, making this work meaningful. Additionally, the datasets used in this study are sourced from previously published works and do not involve any privacy or ethical concerns.

## Acknowledgements

This work was supported by the National Key R&D Program of China (2022YFB3103700, 2022YFB3103704), the National Natural Science Foundation of China (NSFC) under Grants No. 62276248, and the Youth Innovation Promotion Association CAS under Grants No. 2023111.

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

## A  Algorithm of LLMDet

For the detailed implementation process of LLMDetd, please refer to the pseudocode provided below. Algorithm 1 is a dictionary construction algorithm that is completed offline by us or provided to the model holder independently of external systems. Algorithm 2 will be provided to users as a third-party tool.

---

**Algorithm 1:** Dictionary Construction

**Input** : A prompt dataset $P$
A large language model $M$
**Output:** A dictionary $D$ for $M$

```
// Step1: Generate text samples
```
**Procedure** *GenerationText(M, P)*
 `// T is a generation text set`
 $T \leftarrow \emptyset$ ;
 **for** *x in P* **do**
  `// Use M to generate text`
  $t \leftarrow M.\text{generate}(x)$ ;
  $T.\text{append}(t)$ ;
 **end**
 **return** $T$

```
// Step2: Word frequency statistic
```
**Procedure** *WordStatistic(T, n)*
 `// Do counter for text sequence`
 `   to get top-k n-gram`
 $n\text{-gram} \leftarrow \text{CountNgram}(T, n, k)$ ;
 **return** $n\text{-gram}$

```
// Step3: Next token probability
   sampling
```
**Procedure** *NextTokenSampling(M, P, K)*
 `// D_M stores information for`
 `   model M`
 $D_M \leftarrow \emptyset$ is empty list;
 $T \leftarrow \text{GenerationText}(M, P)$ ;
 **for** $n$ *in* $\{2, 3, 4\}$ **do**
  $D_n$ is an empty dictionary ;
  $n\text{-gram} \leftarrow \text{WordStatistic}(T, n)$ ;
  **for** *s in n-gram* **do**
   $p^w \leftarrow M.\text{next\_token}(s_{[1:n-1]})$ ;
   $D_n.\text{add}(\{s_{[1:n-1]} : p^w_{[1:K]}\})$ ;
  **end**
  $D_M.\text{append}(D_n)$ ;
 **end**
 **return** $D_M$

---

**Algorithm 2:** Text Detection

**Input** : A piece of text $t$ for detecting
A list $D = [D_{M_0}, \ldots, D_{M_c}]$ for $c$
LLMs and $D_{M_0}$ denotes human
**Output:** A detection result $R$

```
// Step4: Proxy perplexity
   estimation
```
**Procedure** *ProxyPerplexity(t, $D_M$)*
 `// Ngram () generate one text`
 `   span in t with length n`
 **for** *s, n in* Ngram *(t)* **do**
  Get $D_n$ from $D_M$ ;
  **if** $s_{[1:n-1]}$ *in* $D_n$ **then**
   $p \leftarrow -\log(D_n.\text{index}(s_{[1:n-1]}))$;
   Proxy_PPL += $p$ ;
  **end**
 **end**
 **return** Proxy_PPL

```
// Step5: Result ranking
```
**Procedure** *Rank(t, D)*
 **for** $i$ *in* $\{0, \ldots, c\}$ **do**
  $\text{PPL}_i$
   $\leftarrow \text{ProxyPerplexity}(t, D_{M_i})$;
 **end**
 $p \leftarrow \text{Classifier}([\text{PPL}_0, \ldots, \text{PPL}_c])$ ;
 $\tilde{p} \leftarrow \text{smooth}(p)$ ;
 $\hat{p} \leftarrow \text{softmat}(\tilde{p})$ ;
 $R \leftarrow \text{sort}(\hat{p})$ ;
 **return** $R$

---

## B  LLMDet Using Newer and Larger LLM

In order to explore whether the gap between proxy perplexity (our method) and true perplexity becomes more apparent as the size of LLMs increases, we conduct additional experiments. We replace LLaMA-7B with LLaMA2-13B (Touvron et al., 2023b) while keeping all other experimental settings the same as in the original paper. The detailed experimental results are shown in Table 6 and Table 7.

From the experimental results, when we replace the original LLM with a better-performing and larger-scale LLM, such as replacing LLaMA-7B with LLaMA2-13B, the detection performance remains essentially consistent with the original performance. This indicates that when a better-performing and larger-size LLM is used, the performance gap between proxy perplexity (our method)

Table 6: Experimental results of text detector based on proxy perplexity with LLaMA-7B, and proxy perplexity with LLaMA2-13B.

| Metric (%) | Model Size | Label of Text Source | | | | | | | | |
|---|---|---|---|---|---|---|---|---|---|---|
| | | Human | GPT-2 | OPT | UniLM | LLaMA | BART | T5 | BLOOM | GPT-Neo |
| P ↑ | LLMDet with LLaMA-7B | 98.54 | 76.09 | 79.08 | 90.81 | 95.61 | 97.55 | 86.86 | 84.67 | 84.45 |
| | LLMDet with LLaMA2-13B | 98.45 | 74.67 | 79.97 | 91.47 | 95.70 | 97.46 | 86.64 | 83.15 | 84.60 |
| | True perplexity | 97.97 | 98.54 | 98.25 | 79.02 | 98.54 | 98.94 | 89.77 | 94.41 | 97.09 |
| R ↑ | LLMDet with LLaMA-7B | 99.00 | 78.13 | 73.88 | 91.74 | 97.30 | 98.41 | 87.56 | 83.08 | 83.90 |
| | LLMDet with LLaMA2-13B | 99.04 | 79.07 | 72.64 | 90.53 | 97.49 | 98.11 | 86.99 | 83.15 | 83.98 |
| | True perplexity | 98.99 | 95.92 | 95.70 | 82.25 | 98.46 | 99.73 | 88.44 | 94.29 | 97.61 |
| F1 ↑ | LLMDet with LLaMA-7B | 98.77 | 77.09 | 76.39 | 91.27 | 96.44 | 97.98 | 87.21 | 83.87 | 84.18 |
| | LLMDet with LLaMA2-13B | 98.74 | 76.80 | 76.13 | 91.00 | 96.58 | 97.78 | 86.82 | 83.47 | 84.29 |
| | True perplexity | 98.48 | 97.22 | 96.96 | 80.60 | 98.85 | 99.34 | 89.10 | 94.35 | 97.35 |

Table 7: Comparison of overall performance between text detectors based on proxy perplexity with LLaMA-7B and proxy perplexity with LLaMA2-13B.

| Method | Macro-F1(%) ↑ | R1(ACC)(%) ↑ | R2(%) ↑ | R3(%) ↑ |
|---|---|---|---|---|
| proxy perplexity with LLaMA-7B | 88.14 | 89.51 | 98.00 | 99.64 |
| proxy perplexity with LLaMA2-13B | 87.96 | 89.30 | 98.07 | 99.69 |
| True perplexity | 94.65 | 95.54 | 99.33 | 99.80 |

and true perplexity does not become more obvious.

## C   Additional Analysis for Robustness Testing

From Table 4, it can be observed that as the temperature increases, the accuracy of text generation detection improves.

Regarding this phenomenon, what we need to clarify is that our method calculates proxy perplexity by building a dictionary based on the probability of sampling the next token. When calculating the probability of the next token, we directly use the softmax with a default temperature of 1.0. When the temperature of LLM is set to 1.0, the generated text actually conforms more closely to the probability distribution of the next token in the dictionary we have constructed. At this point, the calculated proxy perplexity is closer to the true perplexity, resulting in higher detection accuracy. Therefore, we can observe that when the temperature is higher, the text distribution generated by the LLM is closer to the probability distribution of the next token in the dictionary, leading to higher detection accuracy.

## D   Sample of $n$-gram for each LLM

Specific examples of 2-gram, 3-gram, and 4-gram for each LLM can be referred to in the tables. Table 8 shows the samples for GPT-2. Table 9 shows the samples for OPT. Table 10 shows the samples for LLaMA. Table 11 shows the samples for T5. Table 12 shows the samples for UniLM. Table 13 shows the samples for BART. Table 14 shows the samples for GPT-Neo. Table 15 shows the samples for Bloom.

Table 8: Samples of $n$-gram for GPT-2.

| 2-gram | 3-gram | 4-gram |
|---|---|---|
| ('not', 'normal') | ('on', 'a', 'robust') | ('to', 'the', 'alleged', 'health') |
| ('rape', 'her') | ('make', 'those', 'customer') | ('on', 'pace', 'for', 'an') |
| ('to', 'agree') | ('mental', 'health', 'clinic') | ('reception', 'at', 'the', 'BBC') |
| ('political', 'campaigning') | ('tower', 'also', 'features') | ('the', 'rule', 'to', 'mean') |
| ('with', 'patients') | ('lost', 'both', 'matches') | ('to', 'get', 'away', 'with') |
| ('the', 'Common') | ('of', 'soccer', '"') | ('she', 'was', 'g', 'ored') |
| ('private', 'men') | ('have', 'been', 'accused') | ('the', 'area', '.]', 'C') |
| ('were', 'victorious') | ('forms', 'of', 'discipline') | ('world', 'economic', 'crisis', ',') |
| ('young', 'team') | ('man', 'who', 'may') | ('to', 'Argentina', ')', 'on') |
| ('said', 'Se') | ('guessing', 'you', 'might') | ('people', 'who', 'are', 'loyal') |
| ('or', 'suspects') | ('the', 'court', 'ruled') | ('partners', '.', 'C', 'G') |
| ('sound', 'energy') | ('service', '.', 'I') | ('our', 'community', 'from', 'this') |
| ('she', 'beat') | ('physical', '.', 'If') | ('train', 'young', 'boys', 'about') |
| ('political', 'career') | ('that', 'Mexicans', 'who') | ('the', 'problem', '.', '"') |
| ('not', 'propose') | ('one', '.', 'We') | ('to', 'Barb', 'ados', 'on') |
| ('today', 'accepted') | ('share', 'of', 'online') | ('them', '.', 'I', "'m") |
| ('was', 'slow') | ('was', 'walking', 'with') | ('of', 'the', 'UEFA', 'presidential') |
| ('receive', 'government') | ('the', 'largest', 'ever') | ('questioned', 'for', 'a', 'week') |
| ('v', 'Arsenal') | ('the', 'shark', ',') | ('think', 'this', 'is', 'incorrect') |
| ('the', 'notes') | ('to', 'have', 'abused') | ('who', 'had', 'been', 'shot') |
| ('smooth', 'and') | ('inquest', 'is', 'now') | ('other', 'animals', 'that', 'have') |
| ('their', 'views') | ('meet', 'President', 'Ts') | ('sale', 'on', 'the', 'market') |
| ('powerful', 'laser') | ('in', 'the', 'procession') | ('struck', 'our', 'city', '.') |
| ('perspective', '"') | ('that', 'is', 'littered') | ('on', 'Sky', 'Sports', '1') |
| ('was', 'introduced') | ('own', 'right', 'which') | ('of', 'the', 'film', 'has') |
| ('said', 'Theresa') | ('justice', 'is', 'fair') | ('that', 'is', 'made', 'for') |
| ('train', '.') | ('it', 'was', 'before') | ('where', 'Richard', 'III', 'was') |
| ('was', 'initially') | ('to', 'get', 'land') | ('teaching', 'children', 'about', '"') |
| ('to', 'rect') | ('has', 'remained', 'silent') | ('winner', 'will', 'receive', 'a') |
| ('now', 'no') | ('match', 'C', '-') | ('the', 'state', 'government', 'dissolved') |
| ('their', 'tuition') | ('in', 'these', 'jobs') | ('the', 'money', ',', 'people') |
| ('seas', 'off') | ('installed', 'with', 'proper') | ('the', 'Irish', 'Cup', 'in') |
| ('play', 'Miss') | ('season', 'at', 'Villa') | ('on', 'the', 'Rights', 'of') |
| ('then', 'punched') | ('same', 'venue', '(') | ('shore', 'line', '.', 'It') |
| ('patients', 'living') | ('when', 'a', 'report') | ('production', 'company', '.', 'C') |
| ('not', 'right') | ('that', 'Mrs', 'Mat') | ('play', 'a', 'movie', ',') |
| ('spin', 'a') | ('great', 'work', 'we') | ('remember', 'that', 'mid', '-') |
| ('usual', '"') | ('the', 'group', 'of') | ('to', 'say', 'that', 'your') |
| ('push', 'their') | ('for', 'all', 'involved') | ('taken', 'it', 'down', '.') |
| ('zoo', 'or') | ('i', 'h', 'lf') | ('that', 'E', 'On', 'continues') |
| ('was', 'talking') | ('on', 'all', 'his') | ('to', 'a', 'harmful', 'retribution') |
| ('obvious', ':') | ('time', '.', 'They') | ('still', 'in', 'college', 'in') |
| ('said', 'Pac') | ('new', "'s", 'ocial') | ('violence', 'in', 'the', 'gay') |
| ('overflow', 'lane') | ('the', 'pub', 'in') | ('with', 'him', '.', 'His') |
| ('the', 'Gibraltar') | ('let', 'out', '"') | ('where', 'they', 'train', 'for') |
| ('showed', 'great') | ('was', 'released', 'on') | ('our', 'finances', '.', 'We') |
| ('that', 'year') | ('the', 'couple', 'with') | ('to', 'ensure', 'their', 'success') |
| ('yours', ',') | ('is', 'at', '."') | ('the', 'stolen', 'car', ',') |
| ('of', 'investments') | ('have', 'obtained', 'a') | ('to', 'prevent', 'future', 'attacks') |
| ('other', 'hints') | ('to', 'let', 'it') | ('our', 'series', 'and', 'will') |

Table 9: Samples of $n$-gram for OPT.

| 2-gram | 3-gram | 4-gram |
|---|---|---|
| ('specific', 'types') | ('from', 'Sam', 'Jones') | ('within', 'the', 'opening', 'few') |
| ('with', 'payments') | ('industry', 'to', 'get') | ('the', 'services', 'of', 'the') |
| ('producer', 'for') | ('then', 'the', 'County') | ('whether', 'the', 'police', 'watchdog') |
| ('was', 'naked') | ('sab', 'er', 'instructor') | ('sport', 'interesting', 'to', 'most') |
| ('place', 'play') | ('the', 'economic', 'impact') | ('shows', 'that', 'there', '"s"') |
| ('visible', 'ethnic') | ('the', 'surgeon', 'actually') | ('racial', 'discrimination', '.', 'The') |
| ('trained', 'by') | ('is', 'a', 'wicked') | ('much', 'better', 'at', 'this') |
| ('placed', 'a') | ('ten', 'years', '.') | ('of', 'leaving', 'the', 'club') |
| ('responsibility', '"') | ('one', 'by', 'a') | ('thread', '.', 'C', 'C') |
| ('situation', 'from') | ('it', 'was', 'Lawrence') | ('ship', 'is', '.', 'I') |
| ('their', 'best') | ('read', 'a', 'particular') | ('old', 'keep', 'is', 'the') |
| ('recommend', 'buying') | ('for', 'one', 'game') | ('protected', 'from', 'over', 'f') |
| ('seen', 'since') | ('very', 'strong', 'turnout') | ('moment', 'the', 'drill', 'sergeant') |
| ('year', 'by') | ('he', 'is', 'a') | ('things', '.', 'And', 'they') |
| ('of', 'Mosul') | ('private', 'college', 'should') | ('started', 'collaborating', 'to', 'solve') |
| ('recent', 'video') | ('then', 'come', 'back') | ('the', 'Guardian', 'last', 'year') |
| ('o', 'lymp') | ('idiot', '.', 'This') | ('other', 'candidates', '.', 'C') |
| ('to', 'Sel') | ('on', 'to', 'her') | ('tourist', 'was', 'missing', ',') |
| ('requests', 'to') | ('pl', 'umes', 'was') | ('outfit', 'for', 'a', 'wedding') |
| ('that', 'four') | ('material', '.', 'He') | ('who', 'Mar', 'low', 'e') |
| ('pattern', '.') | ('doctor', 'prescribed', 'me') | ('seen', 'used', 'on', 'the') |
| ('ultimate', 'opportun') | ('to', 'win', 'promotion') | ('managed', 'to', 'maintain', 'my') |
| ('you', 'if') | ('got', 'no', 'problem') | ('of', 'the', 'London', 'bombings') |
| ('team', 'players') | ('published', 'in', 'February') | ('run', '.', 'The', 'problem') |
| ('quicker', '.') | ('want', 'to', 'move') | ('this', 'movie', 'is', 'complete') |
| ('systems', '"') | ('offence', 'of', '"') | ('the', 'front', 'door', 'to') |
| ('team', 'culture') | ('without', 'undermining', 'the') | ('page', 'of', 'the', 'Hong') |
| ('son', 'away') | ('event', 'on', 'February') | ('vandalism', 'without', 'the', 'people') |
| ('themselves', 'for') | ('not', 'officially', 'recognised') | ('the', 'energy', 'sector', 'in') |
| ('your', 'school') | ('than', 'a', 'typical') | ('natural', 'selection', '.', 'Sometimes') |
| ('winding', 'roads') | ('first', '.', 'If') | ('the', 'same', 'stadium', 'and') |
| ('of', 'Idlib') | ('effort', 'from', 'C') | ('selling', 'information', 'regarding', 'a') |
| ('of', 'seven') | ('progressed', 'through', 'the') | ('limit', ',', 'especially', 'in') |
| ('subsequently', 'charged') | ('the', 'Blues', '.') | ('that', '.', 'I', 'just') |
| ('site', 'AH') | ('tigers', 'in', 'captivity') | ('some', 'sort', ',', 'at') |
| ('perform', 'tasks') | ('gl', 'iding', '.') | ('one', 'is', 'the', 'most') |
| ('to', 'these') | ('this', 'has', 'not') | ('to', 'Deputy', 'First', 'Minister') |
| ('remember', 'who') | ('hosts', 'Japan', ',') | ('lists', 'and', 'the', 'fact') |
| ('participate', 'and') | ('signed', '20', '-') | ('they', '"re"', 'old', 'enough') |
| ('worked', 'because') | ('term', '"', 'less') | ('pay', 'a', 'tax', 'in') |
| ('too', 'if') | ('excessive', 'force', 'against') | ('their', 'fury', 'and', 'carnage') |
| ('otherwise', 'damaged') | ('make', 'the', 'match') | ('were', 'to', 'acknowledge', 'him') |
| ('on', 'Ukrainian') | ('to', 'Europe', ',') | ('now', '.', 'I', 'don') |
| ('smack', 'in') | ('which', 'won', 'the') | ('that', 'the', 'collision', 'was') |
| ('l', '[') | ('think', 'he', '"d"') | ('repeat', 'of', 'the', 'previous') |
| ('owner', 'did') | ('no', 'and', 'they') | ('then', ',', 'of', 'course') |
| ('single', 'run') | ('of', 'glass', 'and') | ('man', 'who', 'beat', 'the') |
| ('was', 'pulling') | ('how', 'his', 'team') | ('of', 'someone', 'on', 'that') |
| ('the', 'influenza') | ('of', 'data', 'that') | ('when', 'the', 'victim', 'was') |
| ('of', 'pressure') | ('friend', 'who', 'suffers') | ('said', 'captain', 'Jamie', 'Stir') |

Table 10: Samples of $n$-gram for LLaMA.

| 2-gram | 3-gram | 4-gram |
|---|---|---|
| ('trees', '.') | ('particular', 'author', '.') | ('which', 'affect', 'millions', 'of') |
| ('tall', 'man') | ('together', 'as', 'a') | ('"', 'person', 'al', '"') |
| ('tax', 'ing') | ('one', 'hand', ',') | ('to', 'put', 'its', 'foot') |
| ('second', 'Local') | ('only', 'make', 'an') | ('type', 'of', 'aircraft', 'that') |
| ('public', 'meeting') | ('there', 'were', 'none') | ('work', 'for', 'a', 'machine') |
| ('when', 'John') | ('when', 'the', 'organization') | ('would', 'mean', 'E', '4') |
| ('to', 'Australian') | ('this', 'situation', ',"') | ('two', 'Jewish', 'women', 'in') |
| ('services', 'during') | ('their', 'compla', 'ints') | ('the', 'Masters', 'in', 'his') |
| ('the', 'IT') | ('professional', 'Hollywood', 'set') | ('to', 'follow', 'the', 'next') |
| ('woman', 'on') | ('the', 'streets', '.') | ('walking', 'along', 'the', 'edge') |
| ('while', 'G') | ('strong', 'squad', ',') | ('will', 'use', 'the', 'E') |
| ('reports', 'make') | ('per', 'day', '.') | ('the', 'injured', 'were', 'children') |
| ('signed', 'Ar') | ('touched', 'as', 'a') | ('the', 'Title', 'VII', 'of') |
| ('workers', 'above') | ('to', 'light', 'after') | ('to', 'Son', 'ar', 'Qu') |
| ('requirement', 'does') | ('tent', 'that', 'he') | ('the', 'resources', 'to', 'respond') |
| ('some', 'thousands') | ('opening', 'took', 'place') | ('volunte', 'ering', 'at', 'her') |
| ('the', 'bigger') | ('play', ',', 'Henry') | ('which', 'are', 'responsible', 'for') |
| ('study', 'but') | ('tie', 'against', 'D') | ('shows', 'ins', 'ur', 'ers') |
| ('then', 'return') | ('officials', 'in', 'Py') | ('to', 'current', 'Health', 'Secretary') |
| ('the', 'purs') | ('of', 'negative', 'economic') | ('way', 'to', 'New', 'Hope') |
| ('the', 'tall') | ('should', 'consider', 'an') | ('to', 'the', 'aircraft', ',') |
| ('remain', 'unclear') | ('year', 'reve', 'als') | ('signs', 'to', 'better', 'represent') |
| ('stronger', 'start') | ('that', 'libraries', 'offer') | ('the', 'Australian', 'Government', 'has') |
| ('that', 'while') | ('on', 'Russia', ',') | ('under', 'ne', 'ath', '.') |
| ('with', 'Niger') | ('the', 'super', 'visor') | ('week', 'in', 'the', 'U') |
| ('she', 'holds') | ('r', 'ides', '.') | ('the', 'trip', 'to', 'S') |
| ('spot', 'where') | ('sector', ',', 'to') | ('shooting', 'on', 'Sunday', 'morning') |
| ('spot', ',"') | ('su', 'cker', 'p') | ('the', 'discovery', 'of', 'the') |
| ('take', 'responsibility') | ('put', 'an', 'end') | ('to', 'a', 'Liberal', 'Dem') |
| ('week', 'since') | ('use', 'the', 'art') | ('to', 'take', 'action', '.') |
| ('what', 'actions') | ('some', 'new', 'people') | ('went', 'wrong', '.', 'I') |
| ('week', 'announced') | ('team', 'call', '-') | ('young', 'woman', 'while', 'she') |
| ('were', 'split') | ('of', 'CR', 'PF') | ('with', 'clubs', 'such', 'as') |
| ('sensitive', '.') | ('province', 'of', 'Gal') | ('total', ',', 'approximately', '') |
| ('the', 'stem') | ('slo', 'ppy', 'at') | ('these', 'suspect', 's', 'are') |
| ('with', 'cred') | ('when', 'she', 'fought') | ('the', 'pool', 'f', 'encing') |
| ('regarding', 'Islam') | ('pay', 'rise', 'they') | ('the', 'process', 'is', 'that') |
| ('this', 'before') | ('to', 'the', 'bands') | ('visitors', 'a', 'year', ',') |
| ('their', 'hands') | ('our', 'operations', 'and') | ('to', 'Gl', 'ouc', 'esters') |
| ('supp', 'lier') | ('underlying', 'coll', 'ater') | ('with', '', '1', '9') |
| ('shoot', 'and') | ('now', ',"', 'she') | ('surv', 'ive', '"', 'and') |
| ('recovery', '.') | ('of', 'deb', 'ts') | ('was', 'still', 'very', 'tight') |
| ('to', 'migr') | ('team', 'that', 'took') | ('tax', 'bill', 'ever', 'handed') |
| ('struggling', 'this') | ('remaining', 'to', 'take') | ('wild', 'f', 'ires', 'that') |
| ('summar', 'ize') | ('of', 'the', 'del') | ('vehicle', 'and', 'a', 'ped') |
| ('society', 'which') | ('the', 'problem', 'before') | ('simple', 'concept', '.', 'Well') |
| ('settlement', 'with') | ('whose', 'client', 'died') | ('to', 'driving', 'an', 'older') |
| ('that', 'fac') | ('yard', 'lines', 'and') | ('which', 'he', 'highlight', 'ed') |
| ('was', 'smart') | ('recip', 'ient', 'will') | ('thr', 'illing', 'cl', 'ash') |
| ('"', 'K') | ('series', ',', 'winning') | ('village', 'of', 'V', 'ran') |

Table 11: Samples of $n$-gram for T5.

| 2-gram | 3-gram | 4-gram |
|---|---|---|
| ('with', 'Bill') | ('very', 'impressed', '.') | ('there', 'is', 'nothing', 'in') |
| ('several', 'bands') | ('with', 'Jamaica', 'n') | ('won', 'the', 'bronze', 'at') |
| ('that', 'recognized') | ('the', 'tour', ',') | ('was', 'what', 'I', 'was') |
| ('tray', 'by') | ('packaged', 'and', 'transported') | ('the', 'Ca', 'ffe', 'N') |
| ('to', '1977') | ('the', 'products', 'and') | ('year', 'to', 'assess', 'the') |
| ('stunning', 'terrace') | ('of', 'assets', 'including') | ('world', '.', 'Land', 's') |
| ('shocked', 'that') | ('to', 'the', 'Super') | ('to', '', 'achieving', 'that') |
| ('purchase', 'Abdul') | ('rural', '.', 'The') | ('their', 'home', 'games', '.') |
| ('right', 'while') | ('they', 'hope', 'to') | ('with', 'him', 'was', 'the') |
| ('training', 'after') | ('our', 'mouth', 's') | ('worth', '', 'mentioning', 'that') |
| ('received', '') | ('then', 'to', 'take') | ('to', 'basic', 'health', 'care') |
| ('respiratory', 'syndrome') | ('support', 'the', 'community') | ('time', '.', 'The', 'company') |
| ('visitors', 'got') | ('one', 'shot', 'on') | ('to', 'win', ',', '"') |
| ('understood', 'the') | ('wedding', 's', ',') | ('would', 'score', 'in', 'the') |
| ('software', 'provides') | ('opposition', 'of', '') | ('with', 'business', 'activity', 'up') |
| ('show', 'any') | ('then', '', 'withdrawn') | ('wrong', 'on', 'U', '.') |
| ('speaks', 'on') | ('out', 'how', 'you') | ('was', 'first', 'introduced', 'in') |
| ('still', 'outstanding') | ('the', 'final', 'spot') | ('that', '', 'a', 'German') |
| ('trial', 'may') | ('per', '100', '.') | ('to', '2003', 'and', 'was') |
| ('their', 'medical') | ('region', ',', 'while') | ('they', 'came', 'into', '') |
| ('use', 'appropriate') | ('writer', 'for', 'iTunes') | ('we', 'have', 'signed', 'the') |
| ('the', 'events') | ('who', 'are', 'assigned') | ('waste', 'must', 'be', 'removed') |
| ('proposed', 'location') | ('the', 'Syrian', 'people') | ('which', 'is', 'funded', 'by') |
| ('way', 'our') | ('sold', 'in', '1986') | ('the', 'British', 'government', 'is') |
| ('working', 'pitches') | ('report', 'his', 'disappear') | ('the', 'anterior', 'anterior', '') |
| ('sixth', 'minutes') | ('which', 'was', 'discovered') | ('very', 'difficult', 'job', ',"') |
| ('thirst', '.') | ('was', 'given', 'bail') | ('was', 'to', 'be', '') |
| ('successor', '.') | ('the', 'experience', 'and') | ('troops', 'are', 'transferred', 'to') |
| ('we', 'all') | ('to', 'the', 'left') | ('what', 'we', 'did', '.') |
| ('want', 'me') | ('under', 'the', 'following') | ('to', 'provide', 'for', 'their') |
| ('returned', 'with') | ('that', 'Mr', 'Cro') | ('the', 'math', 'is', 'only') |
| ('was', 'watch') | ('station', 'in', 'London') | ('track', '.', 'The', 'San') |
| ('their', 'faces') | ('seek', 'to', 'promote') | ('website', '.', 'Mr', 'Burke') |
| ('shot', 'past') | ('office', 'closed', '.') | ('the', 'first', 'lady', 'is') |
| ('trainer', 'Dan') | ('that', 'Walt', 'Disney') | ('was', 'cleared', 'of', 'all') |
| ('the', 'lower') | ('seemed', 'like', 'it') | ('work', 'due', 'to', '') |
| ('the', 'Caribbean') | ('very', 'few', 'people') | ('to', 'use', 'its', 'own') |
| ('wildlife', 'groups') | ('unhealthy', 'fat', 'fat') | ('while', ',', 'but', 'the') |
| ('the', 'Bee') | ('of', 'the', 'heavy') | ('with', 'the', 'Securities', 'and') |
| ('your', 'advanced') | ('strong', 'transport', 'network') | ('that', 'an', 'arrest', 'has') |
| ('winning', 'player') | ('so', 'unfortunately', 'no') | ('was', '', 'largely', 'used') |
| ('successful', '?') | ('"', 'Un', 'der') | ('there', '"', 's', 'more') |
| ('were', 'disappointed') | ('was', 'driving', ',') | ('the', '', 'i', 'Gen') |
| ('why', 'an') | ('project', 'has', 'raised') | ('the', 'best', 'known', 'play') |
| ('represented', 'her') | ('opportunity', 'to', 'help') | ('then', 'charged', '', 'a') |
| ('promote', 'investment') | ('our', 'products', 'and') | ('the', 'drug', 'on', 'the') |
| ('soldier', 'and') | ('plunge', 'in', 'the') | ('them', 'from', 'becoming', 'the') |
| ('started', 'shooting') | ('or', 'delete', 'any') | ('to', 'tell', '.', 'It') |
| ('visiting', 'the') | ('of', 'my', 'way') | ('which', 'ran', 'from', '12') |
| ('questions', '."') | ('raised', 'the', 'limit') | ('that', 'Scotland', 'can', 'win') |

Table 12: Samples of $n$-gram for UniLM.

| 2-gram | 3-gram | 4-gram |
|---|---|---|
| ('our', 'hard') | ('s', 'a', 'ho') | ('minister', 'for', 'immigration', '.') |
| ('the', 'j') | ('f', 'a', '.') | ('will', 'help', 'to', 'di') |
| ('parts', '"') | ('back', 'around', 'the') | ('under', 'p', 'ri', 'vil') |
| ('lead', 'with') | ('window', ',', 'to') | ('was', 'transferred', 'to', 'q') |
| ('na', 'gg') | ('to', 'm', 'east') | ('haze', 'ln', 'ut', 'con') |
| ('meets', 'at') | ('way', 'you', 'would') | ('k', 'ha', 'ka', 'g') |
| ('lit', 'hua') | ('arch', 'er', ',') | ('that', 'm', 'cca', 'be') |
| ('she', 'fell') | ('and', 'legs', ',') | ('on', 'the', 'future', '.') |
| ('third', 'night') | ('hanged', ',', 'and') | ('r', 'aj', 'esh', '.') |
| ('play', 'them') | ('little', 'flu', 'stered') | ('strangers', '"', 'was', 'released') |
| ('oldest', 'and') | ('g', 'ara', 'v') | ('suspended', 'his', 'journalist', 'conduct') |
| ('informal', 'economy') | ('earnings', 'were', 'affected') | ('scored', 'a', 'few', 'goals') |
| ('sword', 'that') | ('round', 'stone', 'co') | ('he', 'was', 'talking', 'about') |
| ('great', 'are') | ('ka', 'wasaki', 'from') | ('parked', 'his', 'car', 'in') |
| ('which', 'features') | ('super', 'sh', 'ow') | ('national', 'team', ',', 'winning') |
| ('responsibility', 'is') | ('modern', '"', 'd') | ('out', 'her', 'j', 'an') |
| ('was', 'convicted') | ('and', 'investment', '.') | ('would', 'see', 'her', 'father') |
| ('grandparents', '.') | ('division', 'was', 'never') | ('say', 'it', '"', 's') |
| ('she', 'gets') | ('facto', '"', 'protector') | ('the', 'mid', '-', '18th') |
| ('showing', 'the') | ('nak', 'a', ',') | ('they', '"', 're', 'spa') |
| ('is', 'really') | ('24', 'de', 'c') | ('in', 'thieves', '"', 'den') |
| ('thank', 'goodness') | ('changes', 'to', 'the') | ('the', 'throat', 'of', 'ka') |
| ('smaller', 'bathroom') | ('new', 'estimate', '.') | ('to', 'the', 'haze', 'ln') |
| ('main', 'reason') | ('g', 'ba', ',') | ('the', 'title', 'is', 'initially') |
| ('not', 'con') | ('her', 'pain', '.') | ('ransom', 'ware', '.', 'it') |
| ('team', ',') | ('her', 'were', 'at') | ('nak', 'hir', 'ni', 'sa') |
| ('my', 'ear') | ('that', 'the', 'hero') | ('of', 'the', 'following', ':') |
| ('parking', 'lots') | ('con', 'ni', 'tor') | ('put', 'pressure', 'on', 'people') |
| ('in', 'terror') | ('that', 'the', 'uk') | ('s', 'a', 'little', 'more') |
| ('n', 'wy') | ('the', 'official', 'w') | ('ho', 'isted', 'itself', 'into') |
| ('re', 'journalists') | ('the', 'crowd', 'was') | ('in', 'their', '49', '-') |
| ('native', 'town') | ('-', 'black', 'boy') | ('sure', 'what', 'he', 'meant') |
| ('protect', '.') | ('most', 'k', 'usa') | ('sa', 'ssa', 'was', 'a') |
| ('happy', 'features') | ('performed', '"', 'm') | ('have', 'us', 'here', 'yet') |
| ('they', 'turned') | ('here', 'and', 'you') | ('hot', ',', '"', 's') |
| ('so', 'd') | ('published', 'in', '1977') | ('the', 'floor', '.', 'm') |
| ('si', 'un') | ('26', '/', '22') | ('he', 'pa', ',', 'and') |
| ('no', 'organised') | ('them', 'as', 'independent') | ('girl', 'who', 'he', 'would') |
| ('u', 'bu') | ('into', 'the', 'water') | ('with', 'a', 'fellow', 'r') |
| ('her', 'album') | ('on', 'plan', 'kt') | ('won', 'every', 'tournament', '.') |
| ('v', 'il') | ('library', 'and', 'archives') | ('verdict', 'was', 'overturned', 'by') |
| ('partner', '"') | ('br', 'id', 'si') | ('see', 'the', 'last', 'one') |
| ('respectively', '.') | ('cafe', 's', ',') | ('order', 'of', 'the', 'ta') |
| ('second', 'fastest') | ('terror', '.', 'the') | ('they', 'moved', 'into', 'a') |
| ('marry', 'men') | ('money', 'for', 'life') | ('saying', ',', '"', 'it') |
| ('my', 'lady') | ('are', 'b', 'on') | ('innocent', 'people', '.', 'o') |
| ('were', 'found') | ('had', 'taken', 'her') | ('would', 'not', 'book', 'the') |
| ('sixth', ',') | ('with', 'a', 'video') | ('the', 'album', '"', 'is') |
| ('fund', 'has') | ('mit', 'e', 'cht') | ('the', 'north', 'k', 'ore') |
| ('years', 'and') | ('you', 'what', '.') | ('had', 'killed', 'her', '.') |

Table 13: Samples of $n$-gram for BART.

| 2-gram | 3-gram | 4-gram |
|---|---|---|
| ('travellers', 'were') | ('search', 'is', 'beginning') | ('who', 'was', 'in', 'the') |
| ('tossed', '.') | ('television', 'News', 'media') | ('up', '"', '......', 'Family') |
| ('targeted', 'online') | ('she', 'told', 'her') | ('them', 'They', 'fought', 'they') |
| ('to', 'tell') | ('team', 'the', 'team') | ('was', 'a', '...', 'Ar') |
| ('two', 'won') | ('the', 'Empire', '"') | ('village', 'today', '!', 'Article') |
| ('was', 'bullied') | ('survive', 'and', 'thrive') | ('will', 'provided', 'master', 'classes') |
| ('transaction', 'details') | ('the', 'reason', 'for') | ('were', 'cited', 'including', ':') |
| ('trouble', 'Officials') | ('would', 'choose', 'this') | ('then', 'he', 'born', 'here') |
| ('victim', 'left') | ('the', 'strong', 'man') | ('with', 'force', ',', 'they') |
| ('use', 'There') | ('G¦', 'Dean', 'Andrews') | ('was', 'jailed', 'in', '2010') |
| ('team', '......') | ('while', 'his', 'comments') | ('victories', 'being', 'scored', '.') |
| ('A', 'School') | ('the', 'report', 'includes') | ('they', 'start', 'again', 'After') |
| ('wrong', '......') | ('were', '.....', 'they') | ('the', 'suspects', '.', 'The') |
| ('worlds', '.') | ('they', 'are', 'working') | ('this', 'youthful', 'group', 'of') |
| ('to', 'abolish') | ('still', 'is', 'a') | ('were', 'false', 'claims', 'that') |
| ('was', 'among') | ('universe', '.', 'The') | ('to', 'explain', 'why', '??') |
| ('to', 'Chad') | ('was', 'sentenced', 'for') | ('train', 'operators', 'The', 'train') |
| ('took', '4') | ('walk', 'out', 'end') | ('the', 'whole', 'series', 'is') |
| ('team', 'Behind') | ('then', 'he', 'and') | ('wishes', 'to', 'remain', 'anonymous') |
| ('visitor', 'st') | ('the', '.', 'Pl') | ('thee', 'to', 'the', 'number') |
| ('that', 'North') | ('who', 'is', 'based') | ('unbeaten', 'and', 'the', 'victory') |
| ('stand', '-') | ('the', 'first', '....') | ('want', 'them', '!', 'They') |
| ('your', 'thing') | ('thousand', 'people', 'missing') | ('years', 'ago', 'The', 'strike') |
| ('the', 'Greek') | ('was', 'hes', 'over') | ('the', 'slot', '...', 'Brian') |
| ('Is', 'My') | ('weather', 'forecast', 'keeps') | ('was', 'at', 'board', '.') |
| ('they', '!!!') | ('singers', 'and', 'designers') | ('which', 'stand', 'Hide', 'Transcript') |
| ('wheel', 'is') | ('will', 'merge', 'it') | ('the', 'university', '.', 'G¦') |
| ('tower', 'house') | ('to', 'saying', 'goodbye') | ('the', 'service', 'that', 'does') |
| ('the', 'dig') | ('.', 'Assistant', 'manager') | ('their', 'website', '|', 'TV') |
| ('verified', '.') | ('through', 'its', 'share') | ('who', 'were', 'sacked', 'in') |
| ('the', 'bastard') | ('selling', 'properties', ',') | ('to', 'passing', 'the', 'mark') |
| ('targeted', 'it') | ('sure', '.', 'Especially') | ('yet', '.......', 'And', '......') |
| ('weeks', 'later') | ('speak', 'is', 'sure') | ('this', 'club', ':', 'Article') |
| ('summer', '.') | ('who', 'was', '82') | ('was', 'likely', 'disappointed', 'in') |
| ('then', 'is') | ('was', 'And', 'That') | ('us', ':', ')', '??') |
| ('the', 'Sam') | ('who', 'came', 'to') | ('they', 'say', ':', 'Turn') |
| ('team', 'is') | ('the', 'race', 'was') | ('who', 'the', 'artist', 'The') |
| ('topic', '-') | ('services', 'are', 'resumed') | ('the', 'south', 'Crew', 'sm') |
| ('spotted', 'nearby') | ('today', ':', 'Jack') | ('with', 'a', 'golf', 'club') |
| ('understandable', 'to') | ('she', 'never', 'heard') | ('G¦', 'The', 'Un', 'ail') |
| ('victim', 'refused') | ('with', 'the', 'age') | ('while', '....', '."', ',', 'as') |
| ('A·', 'advertisement') | ('survives', 'military', 'tests') | ('with', 'women', 'are', 'is') |
| ('strife', 'G¦') | ('to', 'gain', 'insights') | ('trial', 'was', 'strike', 'after') |
| ('trader', 'Bryan') | ('the', 'holidays', 'will') | ('where', 'are', 'you', '?,') |
| ('temples', 'From') | ('space', 'The', 'policy') | ('unavailable', '.', 'for', 'Advertisement') |
| ('surprised', 'and') | ('the', 'items', 'They') | ('was', 'attacked', 'she', 'was') |
| ('widespread', 'as') | ('starts', '!', 'When') | ('G¦', 'The', 'former', '...') |
| ('two', 'titles') | ('union', 'and', 'to') | ('were', 'there', 'were', 'some') |
| ('whose', 'whose') | ('to', 'change', 'A') | ('website', 'said', ':', 'advertisement') |
| ('than', 'live') | ('y', 'rs', '.') | ('then', 'travels', 'C', 'hen') |

Table 14: Samples of $n$-gram for GPT-Neo.

| 2-gram | 3-gram | 4-gram |
|---|---|---|
| ('where', 'about') | ('the', 'bag', 'and') | ('transferred', 'to', 'the', 'Auckland') |
| ('these', 'talks') | ('this', 'context', 'that') | ('told', 'that', 'they', 'had') |
| ('ticket', '.') | ('with', 'Hell', 'fire') | ('l', 'Bul', 'ger', 'G') |
| ('week', ':') | ('to', 'schools', 'that') | ('touch', 'of', 'the', 'night') |
| ('the', 'drones') | ('to', 'anyone', 'in') | ('will', 'be', 'drawn', 'in') |
| ('things', 'happen') | ('then', 'grabbed', 'M') | ('was', 'proposed', 'by', 'Swedish') |
| ('to', 'expose') | ('to', 'sit', 'on') | ('with', 'some', 'describing', 'it') |
| ('the', 'graphene') | ('together', 'on', 'this') | ('why', 'more', 'needs', 'to') |
| ('with', 'girls') | ('the', 'store', 'was') | ('very', 'powerful', 'and', 'personal') |
| ('where', 'residents') | ('l', 'viol', 'ation') | ('ways', 'that', 'participants', 'can') |
| ('their', 'compliance') | ('the', 'early', 'universe') | ('your', 'country', 'and', 'you') |
| ('to', 'getting') | ('with', 'his', 'ability') | ('uphill', 'battle', 'to', 'get') |
| ('then', 'headed') | ('with', 'someone', 'that') | ('whatever', 'I', 'had', 'to') |
| ('undermined', '."') | ('was', 'g', 'usting') | ('universe', 'today', 'is', 'driven') |
| ('unconscious', 'when') | ('told', 'prosecutors', 'that') | ('L', 's', 'first', 'cash') |
| ('the', 'stroke') | ('to', 'persuade', 'S') | ('with', 'men', 'and', 'had') |
| ('works', 'ahead') | ('the', 'end', 'users') | ('L', 's', 'NFL', 'coverage') |
| ('trying', 'several') | ('unique', 'to', 'humans') | ('year', ',', 'after', 'which') |
| ('tourists', 'away') | ('the', 'regeneration', 'of') | ('L', 's', 'other', 'five') |
| ('with', '64') | ('to', 'memor', 'ise') | ('translation', ',', 'the', 'performance') |
| ('was', 'sidelined') | ('very', 'unusual', 'case') | ('will', 'be', 'rolled', 'out') |
| ('to', 'bed') | ('A', '23', 'million') | ('was', 'condemned', 'by', 'the') |
| ('l', 'Secret') | ('time', 'in', 'recent') | ('who', 'have', 'been', 'ravaged') |
| ('l', 'List') | ('they', 'are', 'in') | ('young', 'players', 'GK', 'Wade') |
| ('why', 'not') | ('very', 'sick', 'ly') | ('top', 'five', 'moments', '.') |
| ('to', 'bow') | ('us', ',', 'because') | ('work', '.', 'His', 'anger') |
| ('which', 'already') | ('trans', 'people', 'can') | ('vision', 'to', 'life', ',"') |
| ('with', 'Art') | ('were', 'working', 'towards') | ('toward', 'the', 'port', 'ico') |
| ('twice', 'about') | ('way', 'of', 'narrowing') | ('which', 'saw', 'former', 'owner') |
| ('under', 'arrest') | ('the', 'embassy', "'s") | ('trunk', 'of', 'her', 'car') |
| ('wherever', 'I') | ('their', 'resistance', 'over') | ('violate', 'the', 'UN', 'Charter') |
| ('used', 'Photoshop') | ('AE', '80', 'bn') | ('was', 'hijacked', 'by', 'a') |
| ('the', 'tradition') | ('L', 'they', 'said') | ('was', 'managed', 'by', 'the') |
| ('ward', 'electing') | ('the', 'outcome', 'the') | ('who', 'felt', 'that', 'the') |
| ('zoo', 'then') | ('we', "'re", 'frustrated') | ('using', 'a', 'rapid', 'HIV') |
| ('wrote', 'that') | ('this', 'VIP', 'club') | ('was', 'born', 'in', '1971') |
| ('visa', 'program') | ('L', 'the', 'officer') | ('trying', 'to', 'develop', 'nuclear') |
| ('l', 'stre') | ('L', 's', 'then') | ('were', 'also', 'up', 'by') |
| ('usually', 'accompanied') | ('used', 'by', 'Democratic') | ('us', 'for', 'a', 'new') |
| ('with', 'developers') | ('the', 'slain', 'soldier') | ('was', 'invited', 'to', 'an') |
| ('use', 'ahead') | ('without', 'starting', 'quarterback') | ('l', 'The', 'BCC', 'I') |
| ('wealthy', 'neighborhood') | ('we', 'can', 'talk') | ('was', 'proud', 'to', 'represent') |
| ('L', 'Fil') | ('the', 'heated', 'swimming') | ('was', 'also', 'a', 'featured') |
| ('under', 'this') | ('the', 'cocaine', ',') | ('with', 'your', 'message', 'to') |
| ('was', 'waiting') | ('word', '.', 'He') | ('L', 'as', 'the', 'Boeing') |
| ('l', 'R') | ('to', 'keep', 'using') | ('win', 'over', 'Sporting', '.') |
| ('will', 'forgive') | ('woman', ',', 'I') | ('total', 'assets', ',', 'as') |
| ('the', 'groove') | ('world', 'of', 'acting') | ('L', 'said', 'John', 'F') |
| ('trials', 'in') | ('well', 'as', 'delivering') | ('was', 'the', 'second', 'most') |
| ('turning', 'his') | ('the', 'World', 'Retail') | ('went', 'to', 'Old', 'Trafford') |

Table 15: Samples of $n$-gram for Bloom.

| 2-gram | 3-gram | 4-gram |
|---|---|---|
| ('the', 'tribes') | ('the', 'data', 'optim') | ('they', 'are', 'not', 'legally') |
| ('test', "'s") | ('the', 'police', 'got') | ('to', 'a', 'weak', 'ening') |
| ('various', 'colors') | ('the', 'human', 'system') | ('work', 'men', 'were', 'allowed') |
| ('tickets', 'for') | ('tagged', 'A', 'idan') | ('were', 'also', 'used', 'for') |
| ('than', 'Rp') | ('theory', 'of', 'an') | ('way', 'we', 'think', 'about') |
| ('state', 'might') | ('their', 'recent', 'poor') | ('two', 'appearances', 'for', 'Portugal') |
| ('theatre', 'special') | ('were', 'the', 'Reds') | ('time', 'that', 'they', 'have') |
| ('strategic', 'planning') | ('used', 'for', 'advertising') | ('Gl', 'Women', 'with', 'Phot') |
| ('were', '83') | ('up', 'the', 'Blue') | ('unclear', 'whether', 'or', 'not') |
| ('the', 'General') | ('the', 'same', 'pricing') | ('using', 'this', 'again', '.') |
| ('wrote', 'is') | ('total', 'annual', 'revenue') | ('who', 'is', 'still', 'chairman') |
| ('wife', 'met') | ('tests', 'and', 'normal') | ('tool', 'to', 'educ', 'ate') |
| ('student', 'assaulted') | ('transport', 'ing', 'US') | ('unit', 'in', 'the', 'nation') |
| ('special', 'equipment') | ('surrounding', 'it', '.') | ('trial', 'court', ',', 'he') |
| ('vessel', 'with') | ('suite', '.', 'The') | ('was', 'a', 'handsome', 'young') |
| ('top', 'employers') | ('use', 'in', 'this') | ('think', 'he', 'GL', 'll') |
| ('undoubtedly', 'have') | ('the', 'House', 'Finance') | ('treatment', 'of', 'any', 'individual') |
| ('type', 'at') | ('were', 'arrested', 'by') | ('woman', 'claiming', 'she', 'was') |
| ('the', 'fiscal') | ('staff', '.', 'C') | ('will', 'return', 'from', 'the') |
| ('with', 'obs') | ('was', 'a', 'bank') | ('you', 'can', 'hear', 'me') |
| ('wedding', 'plan') | ('visitors', '.', 'These') | ('use', 'their', 'violence', '.') |
| ('tumor', 'and') | ('the', 'brain', '(') | ('was', 'mentally', 'ill', 'after') |
| ('small', 'fee') | ('with', 'new', 'talent') | ('these', 'materials', '.', 'The') |
| ('unnecessary', 'burden') | ('that', 'is', 'distinguished') | ('to', 'the', 'Detroit', 'Pist') |
| ('yet', 'reviewed') | ('so', 'impressed', 'with') | ('think', 'there', 'is', 'very') |
| ('yarn', 'having') | ('with', 'Jon', 'ah') | ('was', 'then', 'advised', 'that') |
| ('wonder', 'over') | ('technology', 'and', 'financial') | ('which', 'is', 'their', 'honey') |
| ('stop', ';') | ('someone', 'enjoy', 's') | ('which', 'is', 'widely', 'regarded') |
| ('tennis', 'game') | ('up', 'over', '4') | ('to', 'be', 'much', 'of') |
| ('the', 'micro') | ('successful', 'films', 'ever') | ('to', 'write', 'the', 'article') |
| ('three', 'aspects') | ('the', 'American', 'Bar') | ('years', '.', 'Le', 'igh') |
| ('website', ',') | ('will', 'have', 'lots') | ('used', 'to', 'identify', 'new') |
| ('will', 'rise') | ('the', 'pandemic', 'continues') | ('to', 'my', 'daughter', 'GL') |
| ('your', 'C') | ('wave', 'had', 'reached') | ('to', 'worry', 'about', 'my') |
| ('that', 'orph') | ('the', 'opposition', 'play') | ('was', 'a', 'shot', 'from') |
| ('west', '-') | ('statements', 'that', 'are') | ('top', '3', ',', 'and') |
| ('will', 'state') | ('unemployed', 'was', 'around') | ('to', 'do', 'a', 'lot') |
| ('your', 'pocket') | ('to', 'data', 'compiled') | ('weight', 'of', 'the', 'animal') |
| ('the', 'Andes') | ('with', 'a', 'project') | ('to', 'keep', 'our', 'state') |
| ('train', 'had') | ('would', 'not', 'back') | ('varies', 'from', '0', '.') |
| ('trusted', '.') | ('the', 'Summer', 'Olympics') | ('up', 'a', 'new', 'office') |
| ('will', 'know') | ('the', 'British', 'off') | ('they', 'have', 'looked', 'to') |
| ('time', 'varies') | ('the', 'storm', 'had') | ('to', 'fall', 'asleep', 'in') |
| ('training', '(') | ('violence', ',', 'exploitation') | ('vessel', 'which', 'was', 'responsible') |
| ('who', 'ran') | ('was', 'Gl', 'perfect') | ('to', 'the', 'Governor', 'for') |
| ('that', 'treaty') | ('substance', 'use', ',') | ('to', 'the', 'Dallas', 'Cow') |
| ('to', '2014') | ('two', 'human', 'B') | ('variety', 'of', 'insight', 'into') |
| ('store', 'page') | ('to', 'assist', 'Liverpool') | ('video', 'playback', '.', 'The') |
| ('they', 'satisfied') | ('together', 'for', 'nearly') | ('very', 'strong', 'team', '.') |
| ('the', 'effective') | ('well', 'supported', 'by') | ('where', 'we', 'need', 'to') |