# OpenReview forum: "LLMDet: A Third Party Large Language Models Generated Text Detection Tool"
_EMNLP/2023/Conference — EMNLP 2023 Findings_

### Official Review · Reviewer_eKex · 2023-08-01

**Soundness:** 3

**Excitement:**

4: Strong: This paper deepens the understanding of some phenomenon or lowers the barriers to an existing research direction.

**Paper Topic And Main Contributions:**

The paper proposes a new method for machine generated text detection. The method is based on the idea of calculating a proxy "perplexity" based on a corpus of generated N-grams. The authors evaluate the performance of this method and its time efficiency.

**Questions For The Authors:**

- How is the real "perplexity" calculated in RQ1? Is it calculated for the entire text or only for the N-grams?

- How does your method take into consideration that not all the humans have the same writing style? Isn't the fact that all the humans are put into one category a problem?

**Reasons To Accept:**

- The method seems promising and the four constraints that authors defined make sense
- The method is efficient and intuitive

**Reasons To Reject:**

- Missing performance comparison with other approaches. Missing performance comparison on out-of-domain data. *Edit:* This concern seems to be partially addressed during the rebuttal phase and the authors provided some additional baseline results.

- L259 "Perplexity is the probability that the model generates the current sentence". This is not what perplexity is. Eq1 - This does not look like perplexity either, this looks like cross-entropy.

- The writing of the paper seems rushed and there are many grammatical mistakes, unfinished sentences. E.g., in the abstract L016 "tool that can sourcing text", L026 "achieving x3.7 faster for recognizing text". GPT-Zero (or GPT-zero) has footnote with link to FAQ on the first three pages. Etc.

- I don't like the including of GPT-Zero in the efficiency comparison. It is an API that you have no control over. I don't find the comparison fair.

- I don't like the mapping between the four requirements and the four RQs. Especially the first two seems to be forced. How does the fact that the true perplexity can be used for classification says anything about the specificity of your method?

**Reproducibility:**

4: Could mostly reproduce the results, but there may be some variation because of sample variance or minor variations in their interpretation of the protocol or method.

**Reviewer Confidence:**

3: Pretty sure, but there's a chance I missed something. Although I have a good feel for this area in general, I did not carefully check the paper's details, e.g., the math, experimental design, or novelty.

**Typos Grammar Style And Presentation Improvements:**

- Section 3 reiterates what is already said in the Introduction.

- Authors sometimes refer to a tool, sometimes to a method. It should be clear what you want to propose in this paper. The method is interesting enough by its own. The fact that you have it wrapped in a tool somewhere is, at least for me, not that interesting.

- L304, I appreciate the O-complexity for the method, but I don't understand the discussion about the parameters here. Why do we need to know that this is a wrong set of parameters? Why is this even discussed in this Section?

- There are experiment-specific details in Section 4 (parameters, LMs) that should be moved to Section 5

- The LMs used in the experiments are listed at three different places

 - LightGBM should be mentioned in Section 4 already

- I am not sure if the term safety is the correct one for the second constraint (L076). There are many other reason why we might not want to require model's parameters, e.g., they are not accessible.

---

> ### Author Rebuttal · Authors · 2023-08-28
>
> Dear reviewer, we sincerely appreciate your thorough review and valuable feedback. Your agreement with the four proposed requirements regarding the testing is greatly acknowledged. We also understand your concerns and hope our following response addresses your inquiries:
>
> **Q1**: Missing performance comparison with other approaches.
>
> **A1**: In response to this concern, we would like to clarify that the absence of baselines in the paper is due to the fine-grained nature of the detection of LLMDet compared to existing works. Current methods, such as DetecGPT and GPT-Zero, primarily aim to distinguish between human-generated and machine-generated text. In contrast, LLMDet focuses on identifying whether the text was generated by humans or by a specific language model (LLM), making our task more challenging.
>
> To address your concerns, we have conducted additional experiments comparing LLMDet with DetecGPT and GPT-Zero. For the sake of fairness, we categorized all detection results from LLMDet for a specific language model as "machine-generated", without further fine-grained distinction. We use a dataset containing 1,000 text samples for this analysis. Below are the results obtained from these experiments:
>
> | Method              | Accuracy $\uparrow$ (\%) | Time(s) $\downarrow$ | Ratio(GPT-Zero) $\uparrow$ |
> | :---------------- | :------: | :------: | :------: |
> | GPT-Zero        |   86.56   | 2376.87  |  x 1.00 |
> | DetectGPT          |  92.67    | 14354.61 | x 0.17|
> | LLMDet    |  88.18   |  321.90 | x 7.38|
>
> From the experimental results, it is evident that, in terms of detection accuracy, LLMDet outperforms GPT-Zero but lags behind DetectGPT. However, LLMDet demonstrates exceptional capabilities in fine-grained discrimination, as shown in **Table 2 of the paper**, and efficiency—achievements that current existing methods have yet to attain.
>
> **Q2**: Missing performance comparison on out-of-domain data.
>
> **A2**: In order to assess the out-of-distribution (OOD) performance of LLMDet, we introduce an additional experiment. We apply the approach described in this paper to create a novel testing dataset. This dataset consists of 10,000 text samples from various domains, including Wiki-CSAI, Reddit-ELI5, and Open-QA, as well as medical and financial. We employ the LLMDet method for text detection and obtained the following experimental outcomes:
>
> | F1-micro  $\uparrow$ (\%)            | R1(Accuracy) $\uparrow$ (\%) |   R2 $\uparrow$ (\%)  |  R3 $\uparrow$ (\%) |
> | :---------------- | :------: |  :------: |  :------: |
> |     82.00    |   83.36   |  93.78  | 97.33  |
>
> Based on the experimental results, and considering the outcomes in terms of both F1-micro and R1, we observe that LLMDet achieves an out-of-domain (OOD) test accuracy of approximately 82%. Additionally, the metrics for R2 and R3 both exceed 90%, indicating that LLMDet's predictions are highly accurate for the top 2 or 3 labels, often including the correct answers. As a result, our approach exhibits commendable performance in OOD testing.
>
> **Q3**: L259 "Perplexity is the probability that the model generates the current sentence". This is not what perplexity is. Eq1 - This does not look like perplexity either, this looks like cross-entropy.
>
> **A3**: We apologize for the lack of clarity in our initial explanation. Perplexity is not the probability of generating a sentence; rather, it is derived from the negative logarithm of the probability. In Equation 1, what is represented is a form of cross-entropy, which is essentially the logarithmic version of perplexity. The measure we define here is based on an n-gram grammar, and therefore, we refer to it as 'proxy perplexity.'
>
> We chose to use cross-entropy as a proxy for perplexity for the following reason: Compared to traditional measures of perplexity, cross-entropy is more sensitive to discrepancies between predicted and actual probabilities. It imposes stricter penalties for significant deviations, which is advantageous for tasks like LLMDet that require precise and fine-grained categorization.
>
> **Q4**: The writing of the paper seems rushed and there are many grammatical mistakes and unfinished sentences. E.g., in the abstract L016 "tool that can be sourcing text", L026 "achieving x3.7 faster for recognizing text". GPT-Zero (or GPT-zero) has a footnote with the link to FAQ on the first three pages. Etc.
>
> **A4**: Thank you very much for pointing out the grammar errors. We will make the necessary corrections in the paper.
>
> **Q5**: I don't like the inclusion of GPT-Zero in the efficiency comparison. It is an API that you have no control over. I don't find the comparison fair.
>
> **A5**: We sincerely apologize for any potential confusion that may arise from this. Here, we would like to reiterate that in the evaluation of text generation efficiency, the experimental setup in **Table 4 of the paper** involves DetecGPT and True-PPL running on a V100-32G GPU. In contrast, GPT-Zero relies on the CPU for API calls, which may additionally be supported by server-side hardware behind it. LLMDet, on the other hand, performs calculations directly on the CPU.
>
> It is observed that LLMDet required the lowest computational resources. Thus, in terms of efficiency comparison, there is no unfair comparison between LLMDet and GPT-Zero.
>
> **Q6**: I don't like the mapping between the four requirements and the four RQs. Especially the first two seem to be forced. How does the fact that the true perplexity can be used for classification say anything about the specificity of your method?
>
> **A6**: We sincerely apologize for any confusion caused and would like to clarify the following points: The motivation behind our work is guided by four key requirements—Specificity, Safety, Efficiency, and Extendibility. The term "four RQs" refers to the research questions we formulate based on these requirements. These questions also outline the issues that our method, LLMDet, aims to address.
>
> When a detection method exhibits specificity, it signifies that the method can achieve refined differentiation, going beyond the simple categorization of human and machine-generated text. Given that our approach falls under the category of perplexity-based methods, the first research question(RQ) is tailored to such methods. Consequently, we use true perplexity to assess whether this category of methods indeed exhibits Specificity.
>
> The second RQ focuses on the safety aspects of LLMDet. Since users can deploy LLMDet for text detection without requiring direct access to the model, LLMDet can, to some extent, ensure that the model remains resistant to external attacks.
>
> **Q7**: How is the real "perplexity" calculated in RQ1? Is it calculated for the entire text or only for the N-grams?
>
> **A7**: The real "perplexity" in RQ1 is calculated by the equation $PPL(X) = exp \lbrace -\frac{1}{t} \sum_{i}^t \log p_ {\theta}(x_{i} | x_{<i})\rbrace$. It calculates for the entire text.
>
> **Q8**: How does your method take into consideration that not all humans have the same writing style? Isn't the fact that all humans are put into one category a problem?
>
> **A8**: Distinguishing between types of human-generated text at a finer granularity is certainly a topic worthy of thorough discussion. However, LLMDet is primarily focused on specialized human texts and does not delve into more nuanced distinctions. Moreover, the finer differentiation of human texts, whether based on educational level, nationality, or age, remains an open topic for discussion. This is an area we may consider exploring in future research endeavors.
>
> **Q9**：Authors sometimes refer to a tool, sometimes to a method. It should be clear what you want to propose in this paper. The method is interesting enough on its own. The fact that you have it wrapped in a tool somewhere is, at least for me, not that interesting.
>
> **A9**: We sincerely apologize for any inconvenience that the wording of the paper may have caused you. Our original intention is to convey that the 'method' refers to the implementation process of LLMDet, whereas the 'tool' pertains to the detection and utilization processes of LLMDet. We chose to encapsulate LLMDet into a tool to enhance user convenience.
>
> **Q10**: L304, I appreciate the O-complexity for the method, but I don't understand the discussion about the parameters here. Why do we need to know that this is a wrong set of parameters? Why is this even discussed in this Section?
>
> **A10**: The complexity here mainly pertains to the efficiency of storage for the dictionary constructed by LLMDet. Calculations of these parameters demonstrate that LLMDet is efficient in terms of storage.
>
> **Q11**: I am not sure if the term safety is the correct one for the second constraint (L076). There are many other reasons why we might not want to require parameters of the model, e.g., they are not accessible.
>
> **A11**: The concept of safety that we discuss in this context pertains to enabling users to operate LLMDet without needing access to the language model parameters. This feature not only enhances user convenience but also adds a layer of security to the model. While we acknowledge that there could be various other reasons to avoid requiring access to model parameters, such as their potential inaccessibility.
>
> Furthermore, we greatly appreciate the constructive feedback you provided on the weaknesses in our writing. Rest assured, we will make the necessary revisions to the paper based on your insightful suggestions.
>
> We hope this response adequately addresses your concerns.
>
> Best wishes!

---

### Official Review · Reviewer_d37J · 2023-08-04

**Soundness:** 3

**Excitement:**

3: Ambivalent: It has merits (e.g., it reports state-of-the-art results, the idea is nice), but there are key weaknesses (e.g., it describes incremental work), and it can significantly benefit from another round of revision. However, I won't object to accepting it if my co-reviewers champion it.

**Paper Topic And Main Contributions:**

This paper proposes a method to estimate the perplexity of a text generated by a large language model. The authors record the next-token probabilities of salient n-grams from the training text generated by this model. They demonstrate that the proposed method satisfies four capabilities, including specificity, safety, efficiency, and extendibility.

**Questions For The Authors:**

- Question A: Why is the human-generated text collected from another source (lines 424-427 in the paper)?

**Reasons To Accept:**

- This paper conducted extensive experiments on 8 large language models and demonstrated that the proxy perplexity is comparable to the true perplexity.
- The proposed method runs much faster than previous works.


**Reasons To Reject:**

- Although the proposed method runs faster than other detectors, the performance comparison is missing. Therefore, I could not evaluate the difficulty of this task.
- An attacker can also estimate the perplexity of the text and encourage the model to generate text with similar perplexity to human text. Such kind of adaptive attack should be mentioned.


**Reproducibility:**

4: Could mostly reproduce the results, but there may be some variation because of sample variance or minor variations in their interpretation of the protocol or method.

**Reviewer Confidence:**

4: Quite sure. I tried to check the important points carefully. It's unlikely, though conceivable, that I missed something that should affect my ratings.

**Typos Grammar Style And Presentation Improvements:**

- Line 587: ‘n-gram .’ should be ‘n-gram.’

---

> ### Author Rebuttal · Authors · 2023-08-28
>
> Dear reviewer, thank you for your recognition and valuable feedback. We sincerely appreciate your positive evaluation of the effectiveness and efficiency of our approach. We also understand your concerns, and we would like to address them as follows:
>
> **Q1**: Although the proposed method runs faster than other detectors, the performance comparison is missing. Here, are you concerned that it might be our lack of baselines to demonstrate the effectiveness of our method and the difficulty of the task?
>
> **A1**: In response to this concern, we would like to clarify that the absence of baselines in the paper is due to the fine-grained nature of the detection of LLMDet compared to existing works. Current methods, such as DetecGPT and GPT-Zero, primarily aim to distinguish between human-generated and machine-generated text. In contrast, LLMDet focuses on identifying whether the text was generated by humans or by a specific language model (LLM), making our task more challenging.
>
> To address your concerns, we have conducted additional experiments comparing LLMDet with DetecGPT and GPT-Zero. For the sake of fairness, we categorized all detection results from LLMDet for a specific language model as "machine-generated", without further fine-grained distinction. We use a dataset containing 1,000 text samples for this analysis. Below are the results obtained from these experiments:
>
> | Method              | Accuracy $\uparrow$ (\%) | Time(s) $\downarrow$ | Ratio(GPT-Zero) $\uparrow$ |
> | :---------------- | :------: | :------: | :------: |
> | GPT-Zero        |   86.56   | 2376.87  |  x 1.00 |
> | DetectGPT          |  92.67    | 14354.61 | x 0.17|
> | LLMDet    |  88.18   |  321.90 | x 7.38|
>
> From the experimental results, it is evident that, in terms of detection accuracy, LLMDet outperforms GPT-Zero but lags behind DetectGPT. However, LLMDet demonstrates exceptional capabilities in fine-grained discrimination, as shown in **Table 2 of the paper**, and efficiency—achievements that current existing methods have yet to attain.
>
>
> **Q2**: An attacker can also estimate the perplexity of the text and encourage the model to generate text with similar perplexity to human text. Such kind of adaptive attack should be mentioned. Therefore, are you perhaps more concerned about the robustness of LLMDet?
>
> **A2**: Yes, we are indeed focused on the robustness of LLMDet. Concerning the adaptive attack you mentioned—where the aim is to encourage the model to generate text similar in complexity to human-generated text—we believe this could be achieved via methods like contrastive learning or model fine-tuning. Due to time constraints, however, we have only explored model fine-tuning in our supplementary experiments.
>
> Therefore, we employ the approach outlined in Section 5.1 of this paper to generate 16,000 text instances for the Vicuna model, which is an instruction fine-tuned version of LLaMA. These instances were labeled with corresponding text labels set as 'LLaMA.' Subsequently, this test data was utilized to assess the robustness of LLMDet. The test outcomes are presented below:
>
> | R1(Accuracy) $\uparrow$ (\%) |   R2 $\uparrow$ (\%)   |  R3 $\uparrow$ (\%) |
> | :------: |  :------: |  :------: |
>  |   97.78   |  99.07  | 99.39 |
>
> Based on the above experimental results, we observe that LLMDet exhibits strong robustness against this specific adaptive attack. In future work, we plan to further explore the resilience of LLMDet to other variations of adaptive attack methods.
>
> In addition to adaptive attacks, we also explored the robustness of LLMDet from various other perspectives, such as random deletion and hyperparameter changes.
>
> (1) For random deletion, we use the approach outlined in Section 5.1 of this article to generate 16,000 text instances using LLaMA. We assigned their corresponding text labels as LLaMA. For the generated text, we set the deletion rates at 0.1, 0.3, and 0.5, respectively. After randomly deleting words in the text according to these rates, we obtain corresponding test texts for evaluating the robustness of LLMDet. The experimental results are as follows:
>
> | Deletion Rate | R1(Accuracy) $\uparrow$ (\%) |   R2 $\uparrow$ (\%)   |  R3 $\uparrow$ (\%) |
> | :----------------: | :------: |  :------: |  :------: |
> | 0.1  |   90.06   |  97.07  | 99.52 |
> | 0.3  |   89.31   |  99.12  | 99.53 |
> | 0.5  |   87.80   |  99.53  | 99.82 |
>
> From the experimental results above, the accuracy of LLMDet still remains at around 90% in the face of random deletions. Therefore, this indicates that LLMDet exhibits strong robustness against random deletions.
>
> (2) For hyperparameter changes, we use the approach outlined in Section 5.1 of this article to generate 16,000 text instances using LLaMA at temperatures of 0.1, 0.4, 0.7, and 1.0 respectively, and assign their corresponding text labels as LLaMA. These instances are then used as test data to assess the robustness of LLMDet. The test results are as follows:
>
> | Temperature | R1(Accuracy) $\uparrow$ (\%) |   R2 $\uparrow$ (\%)   |  R3 $\uparrow$ (\%) |
> | :----------------: | :------: |  :------: |  :------: |
> | 0.1  |   91.23   |  97.55  | 99.46 |
> | 0.4  |   92.05   |  97.48  | 99.33 |
> | 0.7  |   93.02   |  97.48  | 99.24 |
> | 1.0  |   94.37   |  98.06  | 99.33 |
>
> Based on the experimental results above, LLMDet achieves a text detection accuracy of over 90% across four different temperature values. This indicates that LLMDet is highly robust to variations in hyperparameters.
>
> **Q3**: Why is the human-generated text collected from another source (lines 424-427 in the paper)?
>
> **A3**: Before addressing this question, we would like to clarify that the categorization of human text is one of the outcomes of our detection process. To enable LLMDet to effectively learn features from human text, it is essential to ensure diversity in the sourcing of that text.
>
> To fulfill this need, we have sought to collect as much human text as possible from various fields. Our review of the HC3 dataset reveals that it includes human text sourced from multiple domains such as Wiki-CSAI, Reddit-ELI5, Open-QA, Medicine, Finance, and others. This meets our criteria for diverse human data. Therefore, we chose to incorporate human text from the HC3 dataset into our own dataset.
>
> Thank you again for your valuable feedback. We hope that our previous response adequately addresses your concerns.
>
> Best wishes!

---

### Official Review · Reviewer_WGR6 · 2023-08-10

**Soundness:** 3

**Excitement:**

3: Ambivalent: It has merits (e.g., it reports state-of-the-art results, the idea is nice), but there are key weaknesses (e.g., it describes incremental work), and it can significantly benefit from another round of revision. However, I won't object to accepting it if my co-reviewers champion it.

**Paper Topic And Main Contributions:**

LLMDet is a third-party detection tool developed to identify the source of generated text from specific large language models (LLMs) such as GPT-2, OPT, LLaMA, and others. The tool records the next-token probabilities of salient n-grams as features to calculate proxy perplexity for each LLM. By jointly analyzing the proxy perplexities of LLMs, the source of the generated text can be determined.

Though calculating perplexity for language models for machine-text detection is not new, the author proposed "Dictionary Construction" to do ensemble averaging, improving the final classification accuracy.

**Reasons To Accept:**

I would like to accept the paper because it's a solid paper, with extended experimental results. It would also be great if the author release their datasets to construct the dictionary, that would be very useful for benchmarking different detection algorithms for the communities.

**Reasons To Reject:**

One problem I see is the claim that LLMDet is faster than other perplexity-based algorithms. I don't think there is sufficient data to support this claim, mainly because for GPT-Zero, DetectGPT and True-PPL are tested on different platform based on Section 5.4 For Efficiency (RQ3). I think you could claim LLMDet is faster because it precomputes the dictionary instead of computing on the fly, but there should not be a number specifically to be "x3.7" based on the reasons above.

Update: This has been fully addressed by the authors in the following response.

**Reproducibility:**

4: Could mostly reproduce the results, but there may be some variation because of sample variance or minor variations in their interpretation of the protocol or method.

**Reviewer Confidence:**

4: Quite sure. I tried to check the important points carefully. It's unlikely, though conceivable, that I missed something that should affect my ratings.

---

> ### Author Rebuttal · Authors · 2023-08-28
>
> Dear reviewer, we sincerely appreciate your thorough review and recognition of our work and experimental results. We completely understand the concerns you've raised, and we hope the following response can address your questions:
>
> **Q**: One problem I see is the claim that LLMDet is faster than other perplexity-based algorithms. I don't think there is sufficient data to support this claim, mainly because GPT-Zero, DetectGPT, and True-PPL are tested on different platforms based on Section 5.4 For Efficiency (RQ3). I think you could claim LLMDet is faster because it precomputes the dictionary instead of computing on the fly, but there should not be a number specifically to be "x3.7" based on the reasons above.
>
> **A**: For any confusion that may have arisen, we sincerely apologize. We would like to clarify that the discussion about 'Efficiency' in Section 5.4 is in terms of the speed and hardware requirements of the detection end.
>
> The creation of the dictionary is a one-time process carried out locally by us, the detection tool provider. During the detection stage, this precomputed dictionary can be utilized directly without any additional computation. As a result, the computational overhead associated with dictionary calculations can be disregarded in the detection stage.
>
> The experimental setup in **Table 4 of the paper** involves DetecGPT and True-PPL running on a V100-32G GPU. In contrast, GPT-Zero relies on the CPU for API calls, which may additionally be supported by server-side hardware behind it. LLMDet, on the other hand, performs calculations directly on the CPU.
>
> Our approach not only has the lowest hardware requirements but also achieves the most advanced computational speed. Consequently, our method is more efficient compared to other methods.
>
> Thank you again for your feedback. We hope the previous response addresses your concerns.
>
> Best wishes!

---

### Official Review · Reviewer_RV13 · 2023-08-11

**Soundness:** 3

**Excitement:**

3: Ambivalent: It has merits (e.g., it reports state-of-the-art results, the idea is nice), but there are key weaknesses (e.g., it describes incremental work), and it can significantly benefit from another round of revision. However, I won't object to accepting it if my co-reviewers champion it.

**Paper Topic And Main Contributions:**

The paper proposes LLMDet, a framework to classify the source of a given text. Unlike traditional machine-generated text detection works, LLMDet aims to clarify which model generated the text. LLMDet adopts a statistics n-gram dictionary to estimate perplexity, which makes it an independent third-party detector. The writers also mention four principles for their model, namely specificity, safety, efficiency, and extendibility. LLMDet well perform on the four aspects. Moreover, the paper includes further discussion on the effect of hyper-parameters.

**Questions For The Authors:**

1. As you mentioned in Section 4.1, constructing a statistics n-gram dictionary needs the next-token probability of the generation model. It constrains the utility of LLMDet only on white-box settings. Why don't you also estimate the next-token probability by statistical methods? Is this because of the limited number of training datasets? Moreover, is there any possibility to do something similar to distillation?
2. In Table 1 and Table 2, it seems stronger models like LLaMA and BART are easier to detect compared to GPT-2 and OPT. It is counterintuitive because stronger models should be more similar to human-written ones, which means harder to discriminate. Do you have any future analysis or explanation for this phenomenon?
3. Since LLMDet is rely on next-token probability, is it robust to some perturbation on text? like paraphrasing or random deletion. Moreover, if the generation model slightly updates weights or changes hyperparameters like temperature, top-p, and top-k, will the model still work well?
4. As there will always be generation models that LLMDet did not learn before, what will happen if we input a text generated by a totally new model? Will it be clarified as human-written? Or do we have some rejection mechanism?

**Reasons To Accept:**

1. LLMDet is a detector framework that can be extendible and transferable to various LLMs.
2. LLMDet is aimed at a higher standard detection task -- clarify which model generated the text.
3. LLMDet is safe and neutral since it is based on an independent third-party statistics n-gram dictionary.
4. The writer design a sufficient analysis of the model and align well to their motivation.

**Reasons To Reject:**

1. Since current dominant LLMs are commercial closed-source models, i.e., OpenAI models, LLMDet could only apply on white-box models (those with access to probability) but are not helpful for black-box settings.
2. Many LLM models can change their probability via different methods, for example, updating parameters (like GPT-4, and some models on Huggingface), changing hyperparameters (like temperature, word penalty, etc., or even fine-tuning on backbone model), and adding noise on prompt or prefix. It is worth considering the robustness of this method in these situations.
3. As it is hard to cover all the LLMs (some of them may be undisclosed), the research question of LLMDet should be viewed as a classification task with out-of-domain classes.

**Reproducibility:**

5: Could easily reproduce the results.

**Reviewer Confidence:**

4: Quite sure. I tried to check the important points carefully. It's unlikely, though conceivable, that I missed something that should affect my ratings.

---

> ### Author Rebuttal · Authors · 2023-08-28
>
> Dear reviewer, thank you for providing us with constructive and valuable feedback on our paper. We appreciate your recognition of the motivation and experimental aspects of our paper, as well as your feedback on improving it. We understand your concerns and are committed to addressing the following issues:
>
> **Q1**: As you mentioned in Section 4.1, constructing a statistics n-gram dictionary needs the next-token probability of the generation model. It constrains the utility of LLMDet only on white-box settings. Why don't you also estimate the next-token probability by statistical methods? Is this because of the limited number of training datasets? Moreover, is there any possibility to do something similar to distillation?
>
> **A1**: Before answering this question, we wish to clarify that the inability to detect text generated by commercial closed-source models is not unique to LLMDet; it is a common limitation of existing detection methods as well. Moreover, our approach is more feasible compared to existing methods because it does not require holding the third-party model. We only need to gather some statistical information about it, which can be provided by closed-source model holders.
>
> We became aware of this limitation during the experimental phase. To mitigate it to some extent, we have considered two possible approaches:
>
> (1) In the process of implementing LLMDet, we offer not only detection capabilities but also an extensible interface for closed-source model owners. Details about this implementation can be found in **Algorithm 1 of Appendix A in the paper**. The extended interface aims to secure the model effectively without compromising the interests of the model owners. Through this approach, we hope to encourage more closed-source model owners to participate and contribute to the continuous improvement of the detection ecosystem of LLMDet.
>
> (2) We have also explored using statistical techniques to estimate the next-token probability in proprietary commercial models. However, due to limited data volume, achieving the anticipated results has been challenging. Additionally, generating a significant amount of statistical data comes with considerable costs. As a result, we have included this approach on our list of future work items.
>
> Furthermore, the distillation method you mentioned is a valuable avenue for future exploration. We will certainly consider it in our future research endeavors.
>
> **Q2**: In Table 1 and Table 2, it seems stronger models like LLaMA and BART are easier to detect compared to GPT-2 and OPT. It is counterintuitive because stronger models should be more similar to human-written ones, which means harder to discriminate. Do you have any future analysis or explanation for this phenomenon?
>
> **A2**: Before addressing this question, we would like to clarify that LLMDet is designed specifically for fine-grained text detection, which fundamentally constitutes a multi-class classification problem. Therefore, lower detection accuracy for a specific category (such as GPT-2 or OPT) does not necessarily indicate difficulty in distinguishing it from human-generated text. Instead, it might stem from the high similarity that these models share with text generated by other models, leading to potential confusion.
>
> The specific experimental results can be observed in the confusion matrix presented in **Figure 2 of the paper**. The lower detection accuracy for GPT-2 and OPT, compared to other categories, arises primarily because these categories are prone to being confused with each other, rather than being difficult to distinguish from human-generated text. Additionally, the level of confusion among other models, such as LLaMA, can also be clearly observed in Figure 2.
>
> **Q3**: Since LLMDet is relying on next-token probability, is it robust to some perturbation on text? like paraphrasing or random deletion. Moreover, if the generation model slightly updates weights or changes hyperparameters like temperature, top-p, and top-k, will the model still work well?
>
> **A3**: Examining the robustness of LLMDet requires taking into account the influence of various factors. Due to time limitations, we only consider three scenarios: random deletion, weight updates, and temperature changes. More detailed robustness testing will be included in future work.
>
> (1) For random deletion, we use the approach outlined in Section 5.1 of this article to generate 16,000 text instances using LLaMA. We assigned their corresponding text labels as LLaMA. For the generated text, we set the deletion rates at 0.1, 0.3, and 0.5, respectively. After randomly deleting words in the text according to these rates, we obtain corresponding test texts for evaluating the robustness of LLMDet. The experimental results are as follows:
>
> | Deletion Rate | R1(Accuracy) $\uparrow$ (\%) |   R2 $\uparrow$ (\%)   |  R3 $\uparrow$ (\%) |
> | :----------------: | :------: |  :------: |  :------: |
> | 0.1  |   90.06   |  97.07  | 99.52 |
> | 0.3  |   89.31   |  99.12  | 99.53 |
> | 0.5  |   87.80   |  99.53  | 99.82 |
>
> From the experimental results above, the accuracy of LLMDet still remains at around 90% in the face of random deletions. Therefore, this indicates that LLMDet exhibits strong robustness against random deletions.
>
> (2) For weight updates, we employ the approach outlined in Section 5.1 of this paper to generate 16,000 text instances using the 7B Vicuna model, an instruction fine-tuned version of LLaMA, and set their corresponding text labels as LLaMA. These instances are then utilized as test data to assess the robustness of LLMDet. The test outcomes are presented below:
>
> | R1(Accuracy) $\uparrow$ (\%) |   R2 $\uparrow$ (\%)   |  R3 $\uparrow$ (\%) |
> | :------: |  :------: |  :------: |
> |   97.78   |  99.07  | 99.39 |
>
> Based on the experimental results above, LLMDet demonstrates a high level of robustness against changes in model parameter weights, at least for this particular configuration.
>
> (3) For hyperparameter changes, we use the approach outlined in Section 5.1 of this article to generate 16,000 text instances using LLaMA at temperatures of 0.1, 0.4, 0.7, and 1.0 respectively, and assign their corresponding text labels as LLaMA. These instances are then used as test data to assess the robustness of LLMDet. The test results are as follows:
>
> | Temperature | R1(Accuracy) $\uparrow$ (\%) |   R2 $\uparrow$ (\%)   |  R3 $\uparrow$ (\%) |
> | :----------------: | :------: |  :------: |  :------: |
> | 0.1  |   91.23   |  97.55  | 99.46 |
> | 0.4  |   92.05   |  97.48  | 99.33 |
> | 0.7  |   93.02   |  97.48  | 99.24 |
> | 1.0  |   94.37   |  98.06  | 99.33 |
>
> Based on the experimental results above, LLMDet achieves a text detection accuracy of over 90% across four different temperature values. This indicates that LLMDet is highly robust to variations in hyperparameters.
>
> In summary, LLMDet exhibits strong robustness against certain types of perturbations in the text, such as random deletions, slight weight updates in the generative model, and adjustments to temperature settings.
>
> **Q4**: As there will always be generation models that LLMDet did not learn before, what will happen if we input a text generated by a totally new model? Will it be clarified as human-written? Or do we have some rejection mechanism?
>
> **A4**: Before addressing this issue, we would like to clarify that the output of LLMDet is not a single definitive label. Instead, it generates a combination of multiple labels, each with an associated confidence level (i.e., the probability of each label being correct). Therefore, if users find results with low confidence levels, it indicates that the predictions are either not reliable or that LLMDet is abstaining from classification.
>
> When inputting a text generated by a totally new model that LLMDet has not previously encountered, the system tends to assign relatively low probabilities to each output label. Consequently, the overall prediction confidence is low. This suggests that the predictions are not reliable or that LLMDet chooses not to classify the input.
>
> Thank you once again for your valuable suggestions. We hope this clarification addresses your concerns.
>
> Best wishes!

---

### Official Review · Reviewer_Nwjg · 2023-08-12

**Soundness:** 3

**Excitement:**

3: Ambivalent: It has merits (e.g., it reports state-of-the-art results, the idea is nice), but there are key weaknesses (e.g., it describes incremental work), and it can significantly benefit from another round of revision. However, I won't object to accepting it if my co-reviewers champion it.

**Paper Topic And Main Contributions:**

This paper proposes a text detection tool that can identify which generation source the text is from. It uses n-gram probability sampled from specific language model to calculate the proxy perplexity of LLMs and use these as feature to train a text classifier.

**Reasons To Accept:**

(1) The classification is more fine-grained. Most of the machine generated text detection problem studies binary classification while in the paper, the ourcome is multi-class and can tell the which LLM the text is from.
(2) The proxy perplexity is efficient to compute than the real perplexity, and requires less resources

**Reasons To Reject:**

1) The result is much worse compared to the using true perplexity. Especially here all the experiments are based on small LLMs, the gap might be more obvious when the size of the LLMs increase.
2) The paper didn't compate to any baselines.

**Reproducibility:**

4: Could mostly reproduce the results, but there may be some variation because of sample variance or minor variations in their interpretation of the protocol or method.

**Reviewer Confidence:**

4: Quite sure. I tried to check the important points carefully. It's unlikely, though conceivable, that I missed something that should affect my ratings.

---

> ### Author Rebuttal · Authors · 2023-08-28
>
> Dear reviewer, thank you for your recognition and valuable suggestions. We appreciate your positive
> feedback on our method and performance. We understand your concerns and would like to address them as follows:
>
> **Q1**: The result is much worse compared to using true perplexity. Especially here all the experiments are based on small LLMs, the gap might be more obvious when the size of the LLMs increases. We believe you are concerned about the effectiveness of LLMDet as the scale of LLM increases, right?
>
> **A1**: We appreciate your concern and would like to clarify that employing the true perplexity method serves as an upper bound for detection effectiveness. However, our approach offers greater robustness. This is because true perplexity is sensitive to changes in each word, while our method is not. Additionally, our approach strives to outperform methods based on true perplexity in terms of both model security and computational efficiency, all while maintaining a high level of detection accuracy.
>
> Regarding the issues you are concerned about, the answer can be found in **Figure 3 of the paper**. The scale of the Language Models (LLMs) used in Figure 3 varies, ranging in size from 110 million (UniLM) to 7 billion (LLaMA). From the experimental results, it is evident that whether we extend our approach to a smaller LLM (110 million) or a larger LLM (7 billion), the overall detection performance consistently hovers around 85%, without any significant downward trend. Consequently, when we incorporate new LLMs, the performance gap compared to the method based on real perplexity won't become significantly wide.
>
> **Q2**: The paper did not compare to any baselines. Here, are you concerned that it might be our lack of a baseline to demonstrate the effectiveness of our method and the difficulty of the task?
>
> **A2**：In response to this concern, we would like to clarify that the absence of baselines in the paper is due to the fine-grained nature of the detection of LLMDet compared to existing works. Current methods, such as DetecGPT and GPT-Zero, primarily aim to distinguish between human-generated and machine-generated text. In contrast, LLMDet focuses on identifying whether the text was generated by humans or by a specific language model (LLM), making our task more challenging.
>
> To address your concerns, we have conducted additional experiments comparing LLMDet with DetecGPT and GPT-Zero. For the sake of fairness, we categorized all detection results from LLMDet for a specific language model as "machine-generated", without further fine-grained distinction. We use a dataset containing 1,000 text samples for this analysis. Below are the results obtained from these experiments:
>
> | Method              | Accuracy $\uparrow$ (\%) | Time(s) $\downarrow$ | Ratio(GPT-Zero) $\uparrow$ |
> | :---------------- | :------: | :------: | :------: |
> | GPT-Zero        |   86.56   | 2376.87  |  x 1.00 |
> | DetectGPT          |  92.67    | 14354.61 | x 0.17|
> | LLMDet    |  88.18   |  321.90 | x 7.38|
>
> From the experimental results, it is evident that, in terms of detection accuracy, LLMDet outperforms GPT-Zero but lags behind DetectGPT. However, LLMDet demonstrates exceptional capabilities in fine-grained discrimination, as shown in **Table 2 of the paper**, and efficiency—achievements that current existing methods have yet to attain.
>
> Thank you once again for your thorough review. We genuinely hope that the response provided above can address your concerns.
>
> Best wishes!

---

### Meta-Review · Area_Chair_NCTN · 2023-09-18

**Recommendation:** 3

**Metareview:**

**quality, clarity, originality**

The work, as further evidenced by the deep discussion by both reviewers and authors, is of great quality and originality. The paper, further than just detecting LLM generated or not, works to identify which LLM generated the text. There are limitations that are highlighted through the reviews that the detection tool requires access to the full model, limiting its application as most commercial models are closed. The question on the perplexity calculation was addressed by authors and further provided updated tables (that should be put in the paper itself) as this provides for more clarity and also shared understanding.

The work would benefit for another round of proof reading as highlighted by a number of reviewers just to fix lagging issues.

**Significance**

The work provides insight not just on detection but also on the LLMs themselves. This I think is significant for the field and the audience of this conference.

**Meta review suggestions**

It would be a good idea for the authors to make further notes on how to get over the limitation of not having access to closed source models. For people who might want to build on your work, where should they look to be able to identify steps that could be taken with black box models? There is a large amount of literature on interpretability (even of black box models), could this be useful?

---

### Decision · Program_Chairs · 2023-10-07

**Decision:**

Accept-Findings

**Comment:**

**quality, clarity, originality**

The work, as further evidenced by the deep discussion by both reviewers and authors, is of great quality and originality. The paper, further than just detecting LLM generated or not, works to identify which LLM generated the text. There are limitations that are highlighted through the reviews that the detection tool requires access to the full model, limiting its application as most commercial models are closed. The question on the perplexity calculation was addressed by authors and further provided updated tables (that should be put in the paper itself) as this provides for more clarity and also shared understanding.

The work would benefit for another round of proof reading as highlighted by a number of reviewers just to fix lagging issues.

**Significance**

The work provides insight not just on detection but also on the LLMs themselves. This I think is significant for the field and the audience of this conference.

**Meta review suggestions**

It would be a good idea for the authors to make further notes on how to get over the limitation of not having access to closed source models. For people who might want to build on your work, where should they look to be able to identify steps that could be taken with black box models? There is a large amount of literature on interpretability (even of black box models), could this be useful?